# The LHX2-OTX2 transcriptional regulatory module controls retinal pigmented epithelium differentiation and underlies genetic risk for age-related macular degeneration

**Mazal Cohen-Gulkar**[1☯], **Ahuvit David**[1☯], **Naama Messika-Gold**[1☯], **Mai Eshel**[1], **Shai Ovadia**[1], **Nitay Zuk-Bar**[1], **Maria Idelson**[2], **Yamit Cohen-Tayar**[1], **Benjamin Reubinoff**[2], **Tamar Ziv**[3], **Meir Shamay**[4], **Ran Elkon**[1]\*, **Ruth Ashery-Padan**[1]\*

**1** Department of Human Molecular Genetics and Biochemistry, Sackler Faculty of Medicine and Sagol School of Neurosciences, Tel Aviv University, Tel Aviv, Israel, **2** The Hadassah Human Embryonic Stem Cell Research Center, The Goldyne Savad Institute of Gene Therapy and Department of Gynecology, Jerusalem, Israel, **3** Smoler Proteomics Center, Lorry I. Lokey Interdisciplinary Center for Life Sciences and Engineering, Faculty of Biology, Technion-Israel Institute of Technology, Haifa, Israel, **4** Daniella Lee Casper Laboratory in Viral Oncology, Azrieli Faculty of Medicine, Bar-Ilan University, Safed, Israel

☯ These authors contributed equally to this work.
\* ranel@tauex.tau.ac.il (RE); ruthash@tauex.tau.ac.il (RAP)

**Data Availability Statement:** The RNA-seq and ChIP-seq datasets generated in this study are

## Abstract

Tissue-specific transcription factors (TFs) control the transcriptome through an association with noncoding regulatory regions (cistromes). Identifying the combination of TFs that dictate specific cell fate, their specific cistromes and examining their involvement in complex human traits remain a major challenge. Here, we focus on the retinal pigmented epithelium (RPE), an essential lineage for retinal development and function and the primary tissue affected in age-related macular degeneration (AMD), a leading cause of blindness. By combining mechanistic findings in stem-cell-derived human RPE, in vivo functional studies in mice and global transcriptomic and proteomic analyses, we revealed that the key developmental TFs LHX2 and OTX2 function together in transcriptional module containing LDB1 and SWI/SNF (BAF) to regulate the RPE transcriptome. Importantly, the intersection between the identified LHX2-OTX2 cistrome with published expression quantitative trait loci, ATAC-seq data from human RPE, and AMD genome-wide association study (GWAS) data, followed by functional validation using a reporter assay, revealed a causal genetic variant that affects AMD risk by altering *TRPM1* expression in the RPE through modulation of LHX2 transcriptional activity on its promoter. Taken together, the reported cistrome of LHX2 and OTX2, the identified downstream genes and interacting co-factors reveal the RPE transcription module and uncover a causal regulatory risk single-nucleotide polymorphism (SNP) in the multifactorial common blinding disease AMD.

available from the GEO, under accession number GSE178166. All other relevant data are within the paper and its Supporting Information files.

**Funding:** RA-P laboratory is supported by grants from the Israel Science Foundation (1128/20), Binational Science Foundation (2013016) and European Union COST program under COST Action CA-18116, ANIRIDIA-NET supported (in part) by grant no. 317652 from the Chief Scientist Office of the Ministry of Health, Israel and the Cancer Biology Research Center, Tel Aviv University. RA-P and RE are supported by the Israel Ministry of Science (3-17557). MC's PhD scholarship is supported by the Claire and Amedee Maratier Institute for the Study of Blindness and Visual Disorders, Sackler Faculty of Medicine, Tel Aviv University, Israel. R.E. is a Faculty Fellow of the Edmond J. Safra Center for Bioinformatics at Tel Aviv University. ME was supported in part by a fellowship from the Edmond J. Safra Center for Bioinformatics at Tel Aviv University. The funders had no role in study design, data collection and analysis, decision to publish, or preparation of the manuscript.

**Competing interests:** The authors have declared that no competing interests exist.

**Abbreviations:** AI, allelic imbalance; AMD, age-related macular degeneration; bHLH-LZ, basic domain helix–loop–helix leucine zipper; CRE, *cis*-regulatory element; DEG, differentially expressed gene; DMEM, Dulbecco's Modified Eagle's Medium; ECM, extracellular matrix; eQTL, expression-quantitative trait loci; FDR, false discovery rate; GSEA, gene-set enrichment analysis; GWAS, genome-wide association study; HD, homeodomain; KD, knockdown; LD, linkage disequilibrium; Mitf, microphthalmia-associated TF; NS, non-silencing; PE, pigmented epithelium; RPE, retinal pigmented epithelium; shRNA, short hairpin RNA; SNP, single-nucleotide polymorphism; TF, transcription factor; TSS, transcription start site.

## Introduction

Age-related macular degeneration (AMD) is the leading cause of irreversible visual impairment in older people, accounting for approximately 50% of legal blindness in western countries. Susceptibility to AMD depends on a combination of genetic components and environmental factors [1,2]. Several genome-wide association studies (GWASs) have been applied to AMD, resulting in the identification of approximately 50 loci that are significantly associated with increased risk for this disease. These AMD-associated loci contain over 1,000 candidate risk single-nucleotide polymorphisms (SNPs). For most of these AMD loci, the causal variants, their mode of action, and the affected target genes are unknown. This is primarily due to our limited understanding of the functional significance of noncoding genomic variants. Deciphering the functional significance of genomic variations in complex diseases, such as AMD, therefore requires identification and study of the regulatory regions that control the differentiation and maintenance of the lineages involved in disease pathology.

The development of AMD involves interactions between several tissue types, with primary roles attributed to cells of the retinal pigmented epithelium (RPE), a monolayered barrier of polarized, pigmented epithelia located between the choriocapillaris and the photoreceptors, which is vital for the development, health, survival, and function of the retinal photoreceptors and the choroid [3,4]. In AMD, RPE dysfunction results in progressive accumulation of deposits, termed drusen, on the basal membrane of the RPE, which ultimately leads to hypoxia and choroidal neovascularization and/or progressive loss of RPE and photoreceptor cells. The acquisition and maintenance of tissue-specific gene expression, including that of the RPE, is accomplished through the combined activity of DNA-binding transcription factors (TFs) and epigenetic factors that together restrict the transcriptional activity to approximately 2% of the genome that carries enhancer function. Mapping tissue-specific *cis*-regulatory elements (CREs) of the RPE and the TFs responsible for these regulatory elements' selection is fundamental for understanding the mechanisms governing acquisition and maintenance of cell-specific transcriptional programs and for resolving the contribution of the CREs to complex diseases, such as AMD.

The RPE originates from the bipotential neuroectoderm of the optic vesicle. Following specification and morphogenesis, the neuroectoderm cells turn into the bilayer optic cup, populated by an outer layer of pigmented epithelium (PE) progenitors and an inner layer of retinal progenitor cells. The RPE gradually acquires its cell-specific phenotype during embryogenesis and postnatal stages, which include the formation of a selective retina–blood barrier, protection from light-induced oxidative toxicity through the pigment granules, expression of the visual-cycle genes required for the recycling of retina cells, and renewal of the outer segment by circadian phagocytosis of the shed discs (reviewed in [5–7]).

Several TFs have been reported to play a role in RPE differentiation and maintenance based on phenotypic analyses of mutant mice. Among these, Otx2 the paired-type homeodomain (HD) TF, is essential for patterning of anterior neural plate and subsequent formation of forebrain, midbrain, and early eye primordia, and is also required later in development for RPE differentiation [8–12]. Otx2 is also maintained in the adult RPE, where it is required for the expression of multiple RPE genes [13–16]. Activity of Otx2 in RPE is attributed to its regulation of and interaction with the microphthalmia-associated TF (Mitf), a basic domain helix–loop–helix leucine zipper (bHLH-LZ) TF that is required for the specification and differentiation of melanocytes and ocular pigmented cells [15–19].

Another important early eye determinant is the Lim HD TF Lhx2, the homolog of the fly Apterous (ap) gene. Lhx2 is required for optic vesicle formation [20], a function that has been attributed to both its regulation of the PE and retinal TFs, Mitf and Vsx2, and its role in the

patterning of the neural primordium by restricting the fate of midline structures [21–24]. In the developing, retina Lhx2, together with its obligatory co-factors the LIM-domain-binding proteins LDB1/2 (dLDB/Chip in the fly), function in regulating of retinal progenitor proliferation and competence [25–28]. The role of Lhx2 in RPE differentiation in the specified optic cup progenitors has not yet been so far directly addressed. Nevertheless, LHX2's importance in human RPE differentiation has been implicated by its identification among 9 RPE core genes, which include OTX2 and MITF, which bear the potential to induce, when mis-expressed, the direct transdifferentiation of human fibroblasts into human RPE-like cells [29]. Currently, however, it is not known how these factors function together in RPE differentiation, including the composition of the transcriptional complexes, the associated *cis*-regulatory sites, the involvement of epigenetic remodelers, and the relevance to retinal disease mechanisms. Here, we studied Lhx2/LHX2 in developing mouse and human RPE generated from stem cells (hES-RPE). Through functional and genomic analyses, we reveal that LHX2 functions upstream and together with OTX2 on shared genomic regions (cistrome) in regulation of target genes. Through this feed-forward regulatory module LHX2-OTX2 control the RPE transcriptional program. Proteomic analyses further exposed co-factors of LHX2 and OTX2 that likely mediate the tissue-specific gene regulation, including LDB1 and the SWI/SNF chromatin remodeling complex. Finally, intersecting the map of the LHX2-OTX2 bound cistrome with published genomic data on AMD revealed a causal noncoding risk-SNP that acts by altering TRPM1 expression in the RPE through the modulation of LHX2 binding to its promoter. The study exemplifies how delineation of tissue-specific transcriptional regulators, their cistromes, and downstream gene-regulatory networks can provide insights into a complex disease's pathology.

## Results

### LHX2 is required for the maintenance of RPE gene expression in vivo and in differentiated human RPE

To directly examine the roles of *Lhx2* in vivo in specified RPE progenitors, we conditionally deleted *Lhx2* from the PE progenitors using the *Lhx2^loxP* line [30] and *Dct-Cre* (previously termed *Tyrp2-Cre* [31]). *Dct-Cre* is active in the specified PE progenitors of the optic cup starting with onset of RPE differentiation around embryonic day 10 (E10) [31–33]. The analyses were conducted on *Lhx2^loxP/loxP; Dct-Cre* (termed *Lhx2*-PE-cKO) mice and control litter mates that do not carry the *Dct-Cre*. At postnatal day 0.5 (P0.5), the PE in the control mice forms a single-layer RPE (Fig 1A). In contrast, the PE of *Lhx2*-PE-cKO mice is non-pigmented and multilayered (Fig 1B). The PE progenitors located in the peripheral optic cup normally differentiate into the pigmented layers of the iris and ciliary body [34]. These anterior structures, including the vitreous, fail to form in the *Lhx2*-PE-cKO mice, further contributing to severe congenital microphthalmia (Fig 1A and 1B).

Distinct morphological and molecular changes were also evident in the *Lhx2*-PE-cKO's embryonic optic cup. While the PE of control mice at E12.5 is organized in a single epithelium layer expressing Lhx2 and P-cadherin (P-cad/Cdh3, Fig 1C), the Lhx2 protein was not detected in the PE of the *Lhx2*-PE-cKO embryos, and the PE was disorganized and failed to express Cdh3 (Fig 1D). The TFs Pax6, Mitf, and Otx2 and cell-adhesion molecule E-cadherin (E-cad/Cdh1), which are normally expressed at this and later stages (Fig 1E, 1G and 1I), all exhibited diminished levels in the *Lhx2*-PE-cKO RPE (Fig 1F and 1H). The loss of Otx2 and Cdh1 persisted in the *Lhx2*-PE-cKO multilayered structure at later stages (E16.5, Fig 1J). In contrast to the aforementioned TFs, Sox9 expression was maintained in the *Lhx2*-PE-cKO RPE, suggesting that its expression is independent of Lhx2 activity (Fig 1K and 1L). We further examined

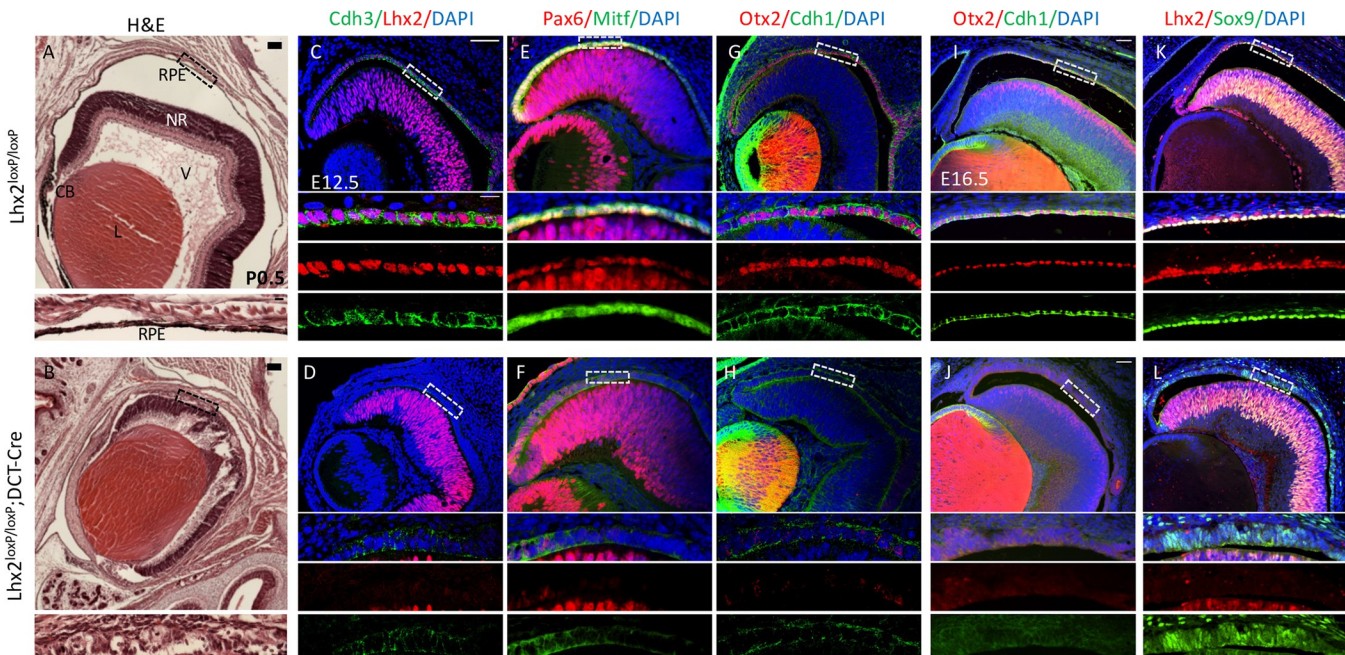

**Fig 1. Lhx2 is required for maintaining RPE fate in mice.** *Lhx2loxP/loxP* control (A, C, E, G, I, K) and *Lhx2loxP/loxP;Dct-Cre* conditional mutation in RPE (B, D, F, H, J, L) analyzed by hematoxylin and eosin (HE) at P0 (A, B) and at E12.5 by antibody labeling for detection of (C, D) Cdh3 (green) and Lhx2 (red); (E, F) Mitf (green), Pax6 (red); (G, H) Cdh1 (green), Otx2 (red), and at E16.5 by antibody labeling for detection of (I, J) Cdh1 (green), Otx2 (red); (K, L) Sox9 (green), Lhx2 (red). Counterstaining with DAPI (blue). The insets below are higher magnifications of the RPE presenting the merged and separate channels. Scale bar in A, B is 100 μm insets are 10 μm. Scale bar C–L is 50 μm, lower insets are 10 μm. CB, ciliary body; HE, hematoxylin and eosin; L, lens; NR, neuroretina; RPE, retinal pigmented epithelium; V, vitreous.

whether the L*hx2*-mutant RPE undergoes a change in cell fate into neural tissue, as previously reported for *Otx2* or *Mitf* mutant mice [12,17]. However, we did not detect mis-expression of Vsx2, Tubb3/Tuj1, or NF-165 in the *Lhx2*-PE-cKO PE (S1 Fig). We therefore conclude that Lhx2 is required for maintaining both the pigmented and neural competence of the PE.

To study the role of LHX2 in differentiated human RPE, we utilized human RPE generated from human embryonic stem cells (hES-RPE). These cells, as RPE from induced pluripotent stem cells, were previously reported to acquire terminal differentiation properties, including morphology, gene expression, and function, based on the rescue of retinal degeneration in animal models and are currently employed in clinical trials for cell-replacement therapies [4,35,36]. hES-RPE partly dedifferentiate following their detachment while splitting, but they do regain differentiation characteristics when grown under differentiation conditions for 14 days (d14). The transition from a dedifferentiated to differentiated phenotype is evident from the reduction in cell size and the acquisition of a polygonal shape, pigmentation, and cell polarity (S2A and S2B Fig). In line with the appearance of terminal differentiation properties in the cells on d14, gene-set enrichment analysis (GSEA) comparing the expression profiles of d5 and d14 hES-RPE (S2C Fig, S1 Table, [33]) showed that genes that were up-regulated on d14 were strongly enriched for RPE signature genes, previously identified to be highly expressed in human RPE [37]. This further demonstrates that hES-RPE presents a valid cellular model for the study of tissue-specific gene-regulation programs in human RPE.

To globally identify the genes regulated by *LHX2* in terminally differentiated human RPE, we performed a knockdown (KD) of *LHX2* using lentiviral-mediated short hairpin RNA (shRNA) transduction in hES-RPE (d14). The lentiviral vector included a GFP for the detection of transduced cells (Fig 2A–2F, green). LHX2, CDH1, and OTX2 were detected in the

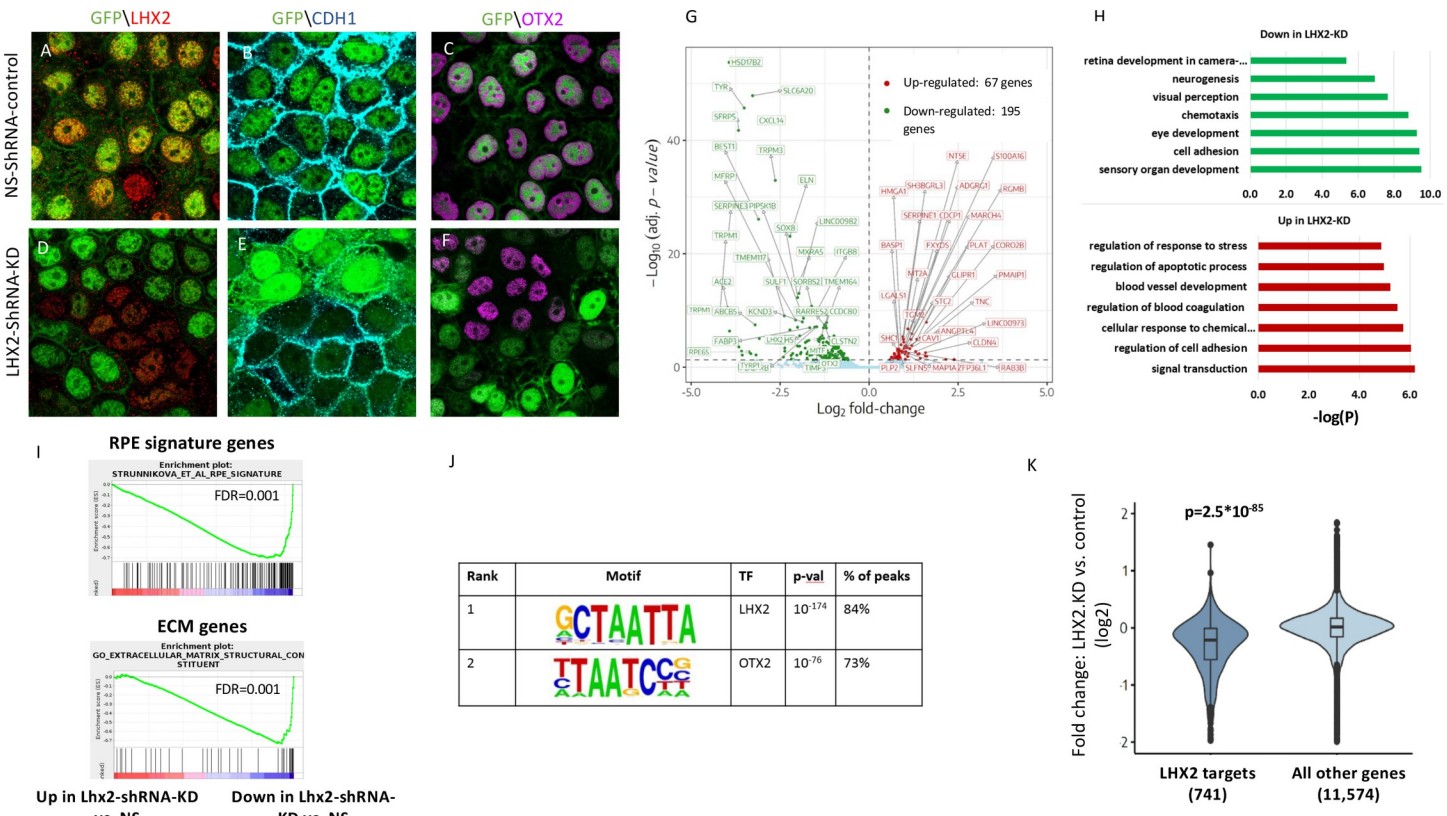

**Fig 2. LHX2 is required for maintaining RPE fate in hES-RPE.** (A–F) Lentiviral transduction of differentiated hES-RPE (d14) with control NS-shRNA (A–C) or LHX2-shRNA (D–F). GFP (green), expressed from the viral vector in the transduced cells. The NS-shRNA (A–C) or LHX2-shRNA (D–F) transduced cells were co-labeled with antibodies to GFP and LHX2 (A, D), CDH1 (B, E), OTX2 (C, F). Bar in A–F is 10 μm. (G) Volcano plot for the differential expression analysis comparing LHX2-KD cells and NS-shRNA hES-RPE control differentiated (d14) hES-RPE cells (differential genes showed |FC|>1.5 and FDR <5%). (H) Enriched GO terms among the down-regulated and up-regulated genes in LHX2 KD differentiated (d14) hES-RPE cells (compared to control differentiated hES-RPE cells transfected by nonspecific shRNAs). (I) GSEA analysis shows that the genes down-regulated upon LHX2 KD in differentiated hES-RPE cells are significantly enriched for RPE signature genes and for ECM genes. (J) De novo motif analysis applied to the 1,278 peaks detected by LHX2 ChIP-seq identified highly significant enrichment for the binding motif of LHX2 and OTX2. (K) The set of putative LHX2 direct target genes as defined by ChIP-seq shows significant down-regulation of expression in hES-RPE cells knocked-down for LHX2. ECM, extracellular matrix; FDR, false discovery rate; GSEA, gene-set enrichment analysis; NS, non-silencing; RPE, retinal pigmented epithelium; shRNA, short hairpin RNA.

hES-RPE transduced with non-silencing (NS) shRNA (Figs 2A–2C and S2 for separate channels), but all of their levels were reduced in the *LHX2*-shRNA-transduced cells (Figs 2D–2F and S2 for separate channels), consistent with the outcome of *Lhx2*-PE-cKO embryonic mouse mutants. To identify the global changes in gene expression between the *LHX2*-KD and control hES-RPE, we performed RNA-seq on the LHX2-KD and control hES-RPE cells. This analysis identified 262 differentially expressed genes (DEGs): 195 down-regulated and 67 up-regulated in *LHX2*-KD cells (|Fold change| > 1.5, *p*-adj <0.05; Fig 2G; S2A Table).

The higher prevalence of down-regulated compared to up-regulated genes following *LHX2* KD supports a role for *LHX2* as a transcriptional activator, in accordance with its mode of action in other lineages, including retinal progenitors and hair follicle stem cells [38,39]. GO enrichment analysis showed that the down-regulated genes were significantly enriched for biological processes associated with cell adhesion ($p = 3.86 \times 10^{-10}$), extracellular matrix (ECM) organization ($p = 3.38 \times 10^{-10}$), and eye development ($p = 5.64 \times 10^{-10}$) (Fig 2H, S3 Table). A comparative GSEA of *LHX2*-KD and control cells, further demonstrated that the down-regulated genes are enriched for the RPE signature [37] and ECM genes (Fig 2I). Key RPE genes that were down-regulated in the *LHX2*-KD cells included *RPE65* (FC = −10.3), *TYR*

(FC = −11.4), *BEST1* (FC = −8.5), *CDH1* (FC = −4.6), *TRPM1* (FC = −18.4), *TRPM3* (FC = −6.2), MITF and OTX2 (both FC = −1.7). Taken together, in both human and mouse RPE, LHX2 is key to maintaining RPE fate, because it regulates the major RPE genes as well as adhesion and ECM proteins important for the epithelial morphology of the RPE.

## Identification and characterization of LHX2-bound enhancers and target genes in human RPE

The genes whose expression in hES-RPE or *Lhx2-PE-cKO* mouse RPE is altered following *LHX2* KD may be direct targets of LHX2 or indirect targets affected by developmental regulators, which are themselves direct targets of LHX2. To identify the direct targets of LHX2 and delineate its gene-regulatory network in the RPE, we performed chromatin immunoprecipitation followed by sequencing (ChIP-seq) using an LHX2 antibody, to systematically map the genomic regions bound by endogenous LHX2 in d14 hES-RPE cells. This ChIP-seq analysis identified 1,278 peaks, which were mapped to 984 genes based on their distance to the nearest transcription start site (TSS) (S4 Table). As expected, de novo motif analysis using Homer [40] showed that the detected peaks were highly enriched for the known LHX2-binding motif (84% of the LHX2 peaks contained an LHX2-binding motif, $p = 1 \times 10^{-174}$; Fig 2J). Notably, the second top-scoring motif was the OTX2-binding motif, detected in 73% of the LHX2 peaks ($p = 1 \times 10^{-76}$; Fig 2J, see below), indicating potential functional cooperation between these 2 TFs in regulation of the RPE transcriptome.

We next integrated the *LHX2*-KD RNA-seq and LHX2 ChIP-seq datasets by linking each peak to its nearest gene's TSS. Of the 984 genes mapped to the LHX2 peaks, 741 were detected in the differentiated hES-RPE cells based on the RNA-seq. Notably, as a group, the expression level of these 741 putative LHX2 target genes was significantly down-regulated upon *LHX2* KD (Wilcoxon's test $p = 2.5 \times 10^{-85}$; Fig 2K). The putative direct targets of LHX2 that were significantly down-regulated upon *LHX2* KD in differentiated hES-RPE cells included genes associated with important functional categories for RPE fate, morphology, and physiology, including cell adhesion, ECM molecules, and ion channels (cadherin: *CDH1*, *PCDH7*; collagens: *COL8A1*, *COL4A3*; integrins: *ITGB8*; laminins: *LAMA2* and ion channels *TRPM1/3*), and key transcriptional regulators (*MITF*, *OTX2*, *SOX5*, *SOX8*; S2A Table). These results strongly indicate that LHX2 is a direct regulator of multiple RPE genes that are required for different aspects of cell fate, morphology, and function.

## Shared direct targets of LHX2 and OTX2 in human RPE

*OTX2* was among the genes identified as putative direct transcriptional targets of LHX2. This is based on its reduced expression following KD of *LHX2* in hES-RPE and in conditional mutant mice (Figs 1 and 2) and on the detection of LHX2-binding sites in 2 CREs associated with the *OTX2* gene (S4 Table, peak 355 and 356).

It was previously reported that OTX2 is required for RPE specification and that it plays a role in photoreceptor survival in mature RPE [12,14,29]. Thus, the down-regulation of OTX2 following *LHX2* KD/cKO in hES-RPE and mouse RPE, respectively, suggests that it could be a key mediator of LHX2 functions in regulating RPE genes. Indeed, the set of OTX2 target genes, previously identified using an *Otx2*-conditional mutation in adult RPE and retinal cells [14], was significantly down-regulated in the *LHX2*-KD RPE cells (Fig 3A, S2B Table).

Importantly, as noted above, motif analysis detected the OTX2-binding site as the second-highest scoring motif in the peaks identified by LHX2 ChIP-seq (Fig 2J). Accordingly, known targets of OTX2 were associated with LHX2-binding sites, suggesting that these 2 TFs co-regulate a shared set of target genes in the RPE. To directly examine the extent of overlap between

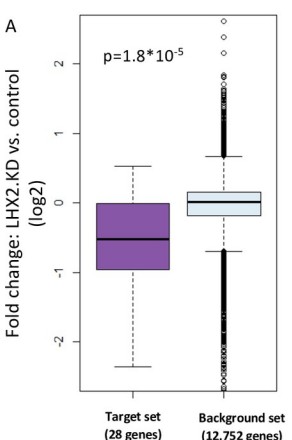

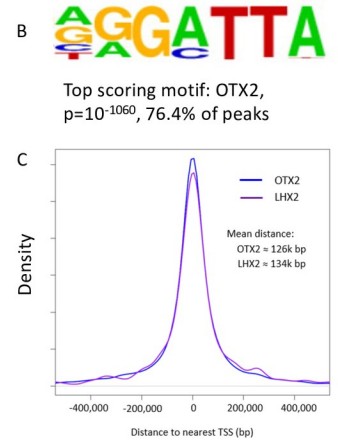

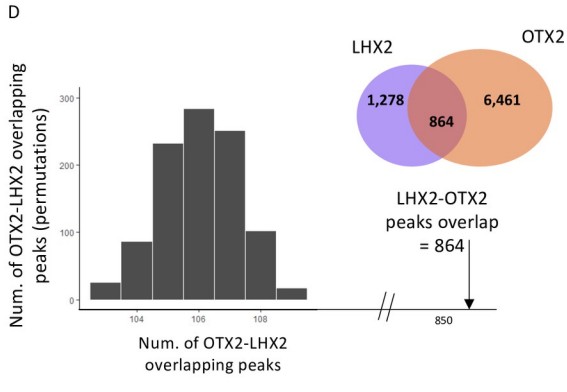

**Fig 3. Shared direct targets of LHX2 and OTX2 in human RPE.** (A) The set of putative Otx2 target genes, previously identified by expression analysis in Otx2-ablated mouse retina, is significantly down-regulated in RPE upon LHX2 KD. The expression analysis in Otx2-ablated retina detected a set of 52 down-regulated genes [14]. We were able to map 41 of these genes to a unique human orthologous gene. Of these, 28 genes were detected in our RNA-seq analysis of RPE cells. This set of 28 genes showed a significant reduction in gene expression in RPE cells knocked-down for LHX2 compared to control cells (p-value calculated using Wilcoxon's test). (B) The top-enriched motif detected by de novo motif analysis on the OTX2 ChIP-seq peaks corresponds to the known OTX2-binding motif. (C) Location distribution of LHX2 and OTX2-binding sites with respect to their nearest TSS. Both TFs show a broad distribution where 50% of the peaks are located >40 kbp from their nearest gene. (D) The observed overlap between LHX2 and OTX2-binding sites (864 overlapping sites) is highly significant, as demonstrated by 1,000 permutations tests in which the average overlap was 105 sites. KD, knockdown; RPE, retinal pigmented epithelium; TF, transcription factor; TSS, transcription start site.

LHX2 and OTX2-binding sites in hES-RPE, we analyzed the genomic regions bound by OTX2 in differentiated hES-RPE cells (d14) using ChIP-seq. The ChIP-seq analysis for OTX2 identified 6,461 peaks, which were mapped to 3,480 genes based on proximity to TSS (S5 Table). De novo motif analysis showed that these genomic regions were highly enriched for the known binding motif of OTX2 (76.4% of the OTX2 peaks contained an OTX2-binding motif, $p = 1 \times 10^{-1060}$; Fig 3B). We next characterized the global distribution of the location of the OTX2 and LHX2 peaks with respect to their nearest TSS. Interestingly, both showed a broad distance distribution. In the case of LHX2, the average distance to its nearest TSS was approximately 134 kbp (median = approximately 46 kbp, Fig 3C), and for OTX2, the average was approximately 126 kbp (median = approximately 43 kbp, Fig 3C). Together, the distribution of the LHX2 and OTX2-binding sites supports a role for both TFs in gene regulation mainly through distal enhancer regions located far from TSSs.

Similar distribution in the genome and the motif enrichments within the bound sites suggested co-occupancy of LHX2 and OTX2. We next intersected the regions bound to LHX2 (1,278) and OTX2 (6,461) and found that 68% of the sites bound by LHX2 were also bound by OTX2 (864 sites; S6 Table). The degree of co-occupancy of these 2 TFs was highly significant ($p < 1 \times 10^{-5}$; based on permutation tests, see Material and methods, Fig 3D). The sites occupied by both TFs showed significant enrichment for both OTX2 and LHX2-binding motifs (binding motifs for both TFs were detected on 582 of these peaks; $p = 1 \times 10^{-103}$ and $p = 1 \times 10^{-98}$, respectively), suggesting cooperative direct chromatin binding of these 2 RPE regulators. The 864 sites co-occupied by both LHX2 and OTX2 were associated with 720 genes, based on proximity to TSS (S6 Table).

Our transcriptomic analysis identified 195 genes that were significantly down-regulated in hES-RPE upon *LHX2* KD. Of these, our ChIP-seq analyses pinpointed 71 (36%) as direct LHX2 target genes and 134 (69%) as direct OTX2 target genes (based on the proximity of the OTX2-binding site to the genes' TSS), including OTX2 itself and several genes that are mutated in inherited forms of retinal degeneration (e.g., RPE65, BEST1, MITF, TYR, MYO7A, and TIMP3) as well as genes associated with elevated risk to AMD (e.g., COL8A1, TRPM1).

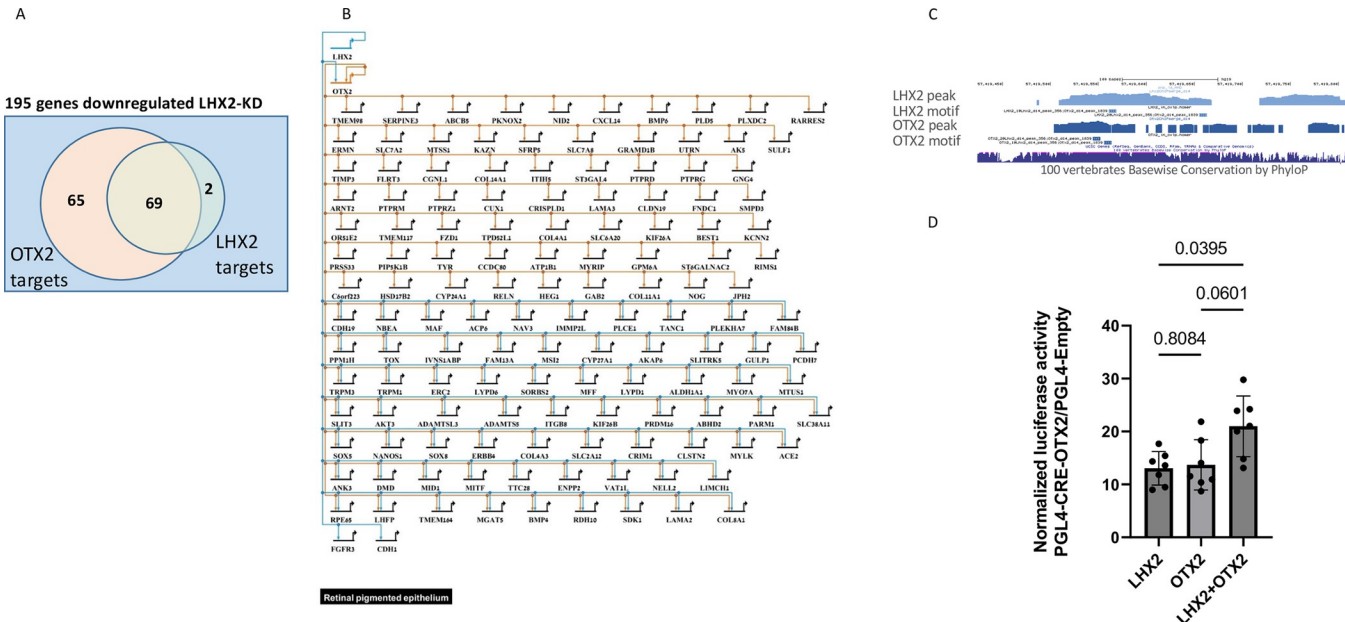

**Fig 4. LHX2–OTX2 gene-regulatory network and functional validation on CRE–OTX2 by reporter assay.** (A) Overlap between OTX2 and LHX2 predicted gene targets based on proximity to the bound regulatory regions and gene down-regulation following LHX2 KD. (B) Model of the LHX2 and OTX2 gene-regulatory networks in hES-RPE cells (generated using BioTapestry; http://www.biotapestry.org/). (C) Overview of a conserved genomic region associated with LHX2 and OTX2 (CRE–OTX2) within the previously identified hs1150 enhancer. The ChIP-seq peaks and motifs of LHX2 and OTX2 are indicated. (D) The PGL4-CRE–OTX2 reporter activity normalized to control PGL4.23-Empty is induced by the co-expression of LHX2 and OTX2 more than the relative elevation by each TF alone. The p-values were calculated by one-way ANOVA, matched, and corrected with Tukey's test for multiple comparisons ($N = 7$, the data and analyses are detailed in S10 Table). CRE, *cis*-regulatory element; KD, knockdown.

Notably, 69 (97%) of these 71 down-regulated direct targets of LHX2 were identified as targets of OTX2, indicating that cooperation with OTX2 is important for transcriptional activity of LHX2 in the RPE (Venn diagram and illustration of gene-regulatory network Fig 4A and 4B, respectively). Down-regulation of the additional 65 OTX2 target genes that were not detected as direct targets of LHX2 can be ascribed to the significant decrease in OTX2 expression observed in the *LHX2*-KD RPE cells.

The detection of 2 LHX2 and OTX2-binding sites upstream of OTX2 support autoregulation of OTX2. Indeed, among these, the enhancer located approximately 150,000 bp upstream of *OTX2* (LHX2_peak_356, Otx2_d14_peak_1839; S6 Table) is contained in a previously reported evolutionarily conserved enhancer element termed hs1150 [41]. This region has been demonstrated to drive expression in a pattern mimicking Otx2, including in the RPE, supporting the functionality of this region in *Otx2* regulation [42]. To examine the transcriptional activity of this genomic region (CRE-OTX2; 1352 bp, chr14:57418450–57419802 (hg19), Fig 4C), we used luciferase reporter assay in D407, a human RPE cell line lacking OTX2 expression [43].

The luciferase reporter activity, normalized to Renilla to account for transfection efficiency, was monitored in response to overexpression of either LHX2 or OTX2, or both (Figs 4D and S3). The normalized luciferase activity of CRE-OTX2 was significantly induced relative to the empty PGL4.23 vector by *LHX2* overexpression (approximately 13-fold $p = 0.012$, $N = 7$) and by *OTX2* overexpression (approximately 14-fold, $p = 0.03$, $N = 7$). The relative reporter activity was further enhanced by the overexpression of both LHX2 and OTX2 (approximately 21-fold, $p = 0.01$, $N = 7$). The enhanced reporter activity following overexpression of both LHX2 and OTX2 as compared to each alone (Fig 4D) further supports their shared transcriptional activity on the CRE-OTX2.

Taken together, our integrated RNA-seq and ChIP-seq analyses reveal, on a genomic scale, shared *cis*-regulatory regions targeted by LHX2 and OTX2, and the feed-forward regulatory hierarchy of these 2 key TFs in maintaining the expression of the RPE genes. The reporter assay further supports the combined function of the 2 TFs in activating a conserved OTX2 enhancer that is likely involved in autoregulation of OTX2 expression in human RPE.

## The LHX2 and OTX2 protein co-factors in hES-RPE

The numerous genomic regions bound by both LHX2 and OTX2 and reduced expression of the genes associated with these regions (based on proximity) suggest that these factors function together on CREs to regulate gene expression in the RPE.

To identify factors interacting with LHX2 and OTX2, we immuno-purified LHX2 or OTX2 from nuclear extracts prepared from hES-RPE (d14) and identified significantly associated polypeptides compared to IgG control using mass spectrometry (IP-MS). In both immunopre-cipitates, we detected strong enrichment of the respective proteins corresponding to the anti-bodies used for the immuno-purification, supporting the specificity of the antibodies (Fig 5A). Moreover, among the proteins interacting with OTX2 was MITF, which was previously reported to interact with OTX2 using pull-down assays in tissue cultures [16,18]. However, we did not detect LHX2 peptides in the OTX2 immunoprecipitate or vice versa. This could be due to low stability of the protein complex and/or to the indirect nature of the interaction between these proteins which might depend on additional co-factors. Interestingly, in both immuno-precipitates, we detected LDB1/2 as well as SSBP2 (single-strand-binding protein 2)—a known co-factor of LDB1 complexes [44,45], detected in the 3 replicates of LHX2 and in 2 of the 3 OTX2 immuno-purified samples (Fig 5A, S9 Table). We further validated the interaction between OTX2 and LDB1 by co-immunoprecipitation of Flag-LDB1 from 293T cells overex-pressing OTX2 or LDB1 alone or together, or in combination with LHX2, followed by western blot analysis with OTX2 antibodies (Fig 5B). The detection of OTX2 in the Flag-LDB1 immu-noprecipitate, with or without LHX2, further supports a direct interaction of human LDB1/2 with OTX2 in hES-RPE. From these results, we infer that LDB1 is directly associated with OTX2 in hES-RPE and that this interaction does not require, and is not inhibited by, LHX2. These findings suggest that LDB1, which is known to interact with LHX2, may mediate the interaction between OTX2 and LHX2, at least on some of the co-bound CREs in the hES-RPE.

In addition to association with LDB1, we detected several SWI/SNF complex subunits in the OTX2 immunoprecipitates, including BRG1/BRM (SMARCA4/2), BAF155/170 (SMARCC1/2), BAF57 (SMARCE1), BAF60a, 60b (SMARCD1/2), and ARID1A/B. Recipro-cally, we detected OTX2 in the BRG1 pulldown, which further supports the physical associa-tion between BRG1 and OTX2 (Fig 5B, S9 Table).

Overall, the results from the IP-MS proteomic analyses suggest that the interaction between LHX2 and OTX2 is mediated through LDB1/2 (Fig 5C). The detection of direct interactions of OTX2 with the BAF complexes suggests a chromatin-remodeling activity for OTX2 and its bound co-factors, allowing robust transcriptional activation of multiple tissue-specific genes (scheme of proposed transcriptional module and co-factors, Fig 5C).

## Identification of an AMD-risk SNP that affects TRPM1 regulation by the LHX2-OTX2 transcriptional complex

We next sought possible links between the enhancers bound by both LHX2 and OTX2 in RPE and genetic predisposition to AMD. Large-scale GWAS studies have identified 35 genomic loci that are significantly associated with AMD risk [46,47]. Like most other complex diseases, for AMD most of the risk variants do not alter protein-coding sequences but rather map to the

A

| IP | LHX2 | | OTX2 | | Brg1 | |
|---|---|---|---|---|---|---|
| Protein | T-test p-value | Log2 fold change | T-test p-value | Log2 fold change | T-test p-value | Log2 fold change |
| LHX2 | 0.015 | 11.0 | | | | |
| LDB1 | 0.013 | 12.6 | 0.0047 | 5.5 | | |
| SSBP2 | 0.019 | 11.0 | 0.1842 | 3.1 | | |
| SMARCA2;SMARCA4 | | | 0.0183 | 6.8 | 0.0139 | 9.0 |
| SMARCC2;SMARCC1 | | | 0.0305 | 5.0 | 0.0037 | 10.5 |
| OTX2 | | | 0.0011 | 16.6 | 0.1842 | 6.3 |
| Mitf | | | 0.0011 | 7.5 | | |
| SMARCD1 | | | 0.0022 | 7.7 | 4.00E-05 | 13.7 |
| SMARCD2 | | | 0.0012 | 8.8 | 0.0001 | 14.3 |
| SMARCE1 | | | 0.0013 | 6.5 | 3.00E-05 | 14.4 |
| ARID1A | | | 0.0802 | 6.6 | 0.0114 | 14.3 |
| ARID1B | | | 0.0741 | 6.1 | 0.0202 | 13.3 |

B

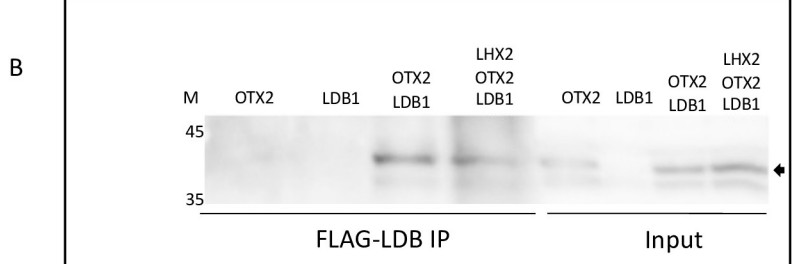

C

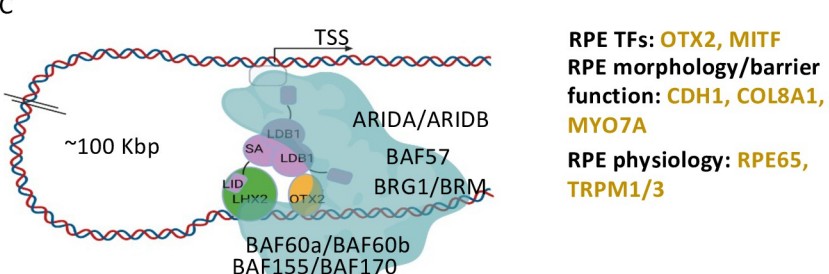

**Fig 5. The LHX2 and OTX2 co-factors in human RPE.** (A) Table summarizing the statistical analysis (*t* test, two-tailed, paired) and the Log2 fold change in proteins bound by LHX2, OTX2, and BRG1 compared to proteins bound by nonspecific IgG based on immunoprecipitation followed by mass spectrometry (IP-MS) analyses. (B) 293T cells overexpressing OTX2, LDB1, LHX2 in the indicated combinations (Input), were subjected to IP with Flag antibodies followed by western blot analyses and immunolabeling with OTX2 antibody, the arrow points to OTX2. S1 Raw image.). (C) Illustration (using Biorander) of the identified co-factors and key downstream targets in hES-RPE. IP, immunoprecipitation; Mitf, microphthalmia-associated TF; RPE, retinal pigmented epithelium; TF, transcription factor; TSS, transcription start site.

noncoding part of the genome, hampering functional interpretation of the GWAS results. To identify causal genes underlying risk for AMD in these genomic loci, Orozco and colleagues recently generated a comprehensive expression-quantitative trait loci (eQTL) resource for the eye [48]. By applying colocalization analysis to GWAS and eQTL signals in each of the 35 AMD-risk loci, that study identified 15 risk loci where, with high confidence, the same genetic variant drives the association with both AMD risk and expression level of the target gene. In these cases of strong colocalization, the eQTL association points to the putative causal AMD-

risk gene in each locus. Interestingly, of the 15 putative causal risk genes for AMD identified by eQTL analysis, the expression level of 3 genes—TRPM1, TSPAN10, and RDH5—is highly enriched in RPE [48], the site of early disease pathology, thereby, specifying these genes as key players in molecular pathways that underlie AMD pathogenesis. We sought to examine possible involvement of LHX2 and OTX2 gene regulation in mediating these AMD-risk signals. Therefore, first, for each of the 15 genomic loci, we generated a set of SNPs associated with AMD-risk (considering in each locus the eQTL tag SNP and all SNPs in high linkage disequilibrium (LD) with it; see Material and methods). Then, we intersected these SNP sets with LHX2 and OTX2 ChIP-seq peaks detected by our analysis. Notably, 1 SNP, rs3809579, which is a significant eQTL for TRPM1 in RPE (Fig 6A) overlaps peaks of both LHX2 and OTX2, located within the TRPM1 promoter, approximately 400 bp upstream of its TSS (Fig 6B). Furthermore, this SNP was located within the LHX2-binding motif, where the C allele fits the consensus while the alternative T allele, which is associated with higher AMD risk, disrupts it (Fig 6C). This SNP is very common in the human population (for example, its minor allele, the C allele, has a frequency of 42% in the European population, based on dbSNP [49]). In accordance with higher LHX2 binding affinity and thus stronger promoter activity for the C allele, eQTL analysis, based on the resource provided by [48] indeed shows that this allele is associated with higher expression level of TRPM1 compared to the T allele (Fig 6A).

To further substantiate that rs3809579 modulates the activity of the regulatory element bound by LHX2 within the TRPM1 promoter, we carried out an allelic imbalance (AI) analysis [50]. In this analysis, only individuals who are heterozygous for the examined SNP are informative. In ChIP-seq data recorded on such individuals, we expect to find in the regulatory element under inspection a markedly higher number of reads originating from the chromosome that carries the allele that confers higher binding affinity than from the other chromosome that carries the allele that reduces binding affinity. However, the hES-RPE cells used in our ChIP-seq experiments are homozygous for this SNP (TT genotype), and thus uninformative for AI analysis of this region. As an alternative resource, we examined the ATAC-seq dataset obtained by Wang and colleagues in human macular RPE [51]. ATAC-seq provides a measure of chromatin openness, which correlates with the activity level of the transcriptional regulatory elements, and thus can be used as a proxy for TF-binding affinity in the AI analysis (Fig 6D). Inspecting ATAC-seq reads covering rs3809579, we identified 2 donors who were heterozygous for this SNP. Reassuringly, for both individuals, the number of ATAC-seq reads that originated from the chromosome with the C allele was 3 times higher than the number of reads originating from the chromosome with the T allele (donor1 – 5T:15C; donor2 – 4T:12C; *p*-value = 0.021), providing support for the notion that the T allele reduces the activity of TRPM1 promoter in the macular region, the affected tissue in AMD (Fig 6E). Last, corroborating TRPM1 as a target gene of LHX2, in our RNA-seq analysis LHX2 KD resulted in an approximately 18-fold reduction in TRPM1 expression levels in RPE (Fig 6F).

To directly examine the effect of the 2 alleles of rs3809579 on the transcriptional activity of this regulatory element within the TRPM1 promoter, we performed a luciferase reporter assay on 311-bp fragment containing either the T or the C alleles (chr15:31394293–31394602, Hg19, denoted CRE-TRPM1-T and CRE-TRPM1-C, respectively). Upon LHX2 overexpression, both constructs induced higher luciferase activity than an empty vector (S4 Fig). Importantly, in accordance with our model, the activation of CRE-TRPM1-C by LHX2 overexpression was significantly higher than the activation of CRE-TRPM1-T (Fig 6G, *p* = 0.01, *N* = 6). These results reveal that rs3809579 not only associates with open chromatin of human RPE (Fig 6E), but also impact the transcriptional output of LHX2 from the TRPM1 promoter. Taken together, our analysis delineates a full causality chain for rs3809579 as a putative causal genetic

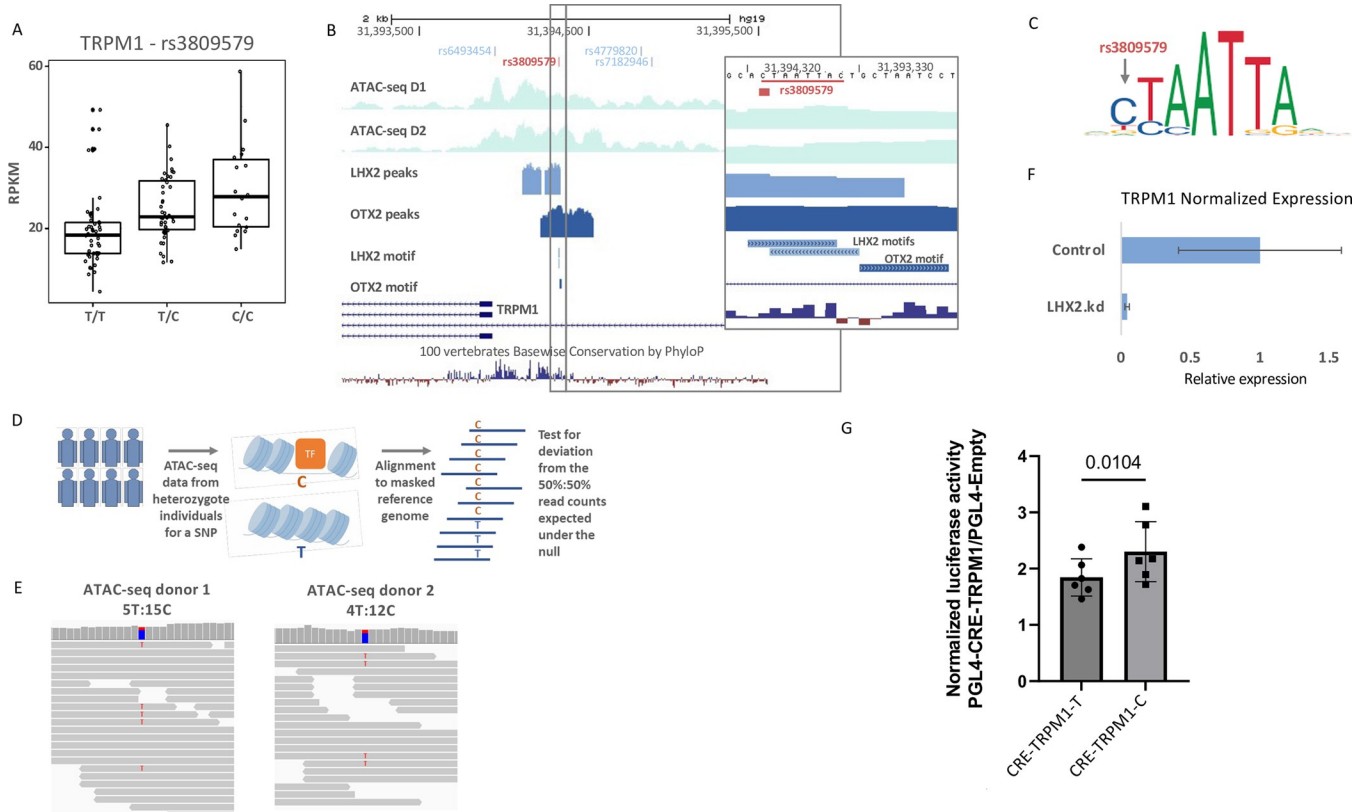

**Fig 6. Identification of causal noncoding risk-SNP of AMD that modulates TRPM1 expression.** (A) rs3809579 is an eQTL for TRPM1 (from http://eye-eqtl.com/; [48]). The C allele is associated with higher expression. (B) An overview of the TRPM1 genomic region including the upstream regulatory regions, the conservation in vertebrates and zoom to the rs3809579 variant (underlie in red), which is located within LHX2 and OTX2 ChIP-seq peaks in the promoter of TRPM1, as well as ATAC-seq tracks from RPE of 2 human donors (D1 and D2). (C) The T allele of rs3809579 disrupts the consensus binding motif of LHX2. The consensus LHX2 PWM is taken from the Jaspar DB [104] (accession: MA0700.1). (D) Illustration of AI analysis, conducted on samples from individuals who are heterozygotes for the examined SNP. The analysis examines significant deviations from the expected 1:1 ratio for counts of reads that originate from the 2 alleles of the examined SNP. Alleles that cause stronger transcriptional activity are expected to be associated with more open chromatin state, and thus, with higher ATAC-seq read counts. (E) RPE ATAC-seq reads from 2 donors (d1, d2) heterozygous for rs3809579 show significantly more reads from the C allele than the alternative T allele. (F) Expression level of TRPM1 is reduced by more than 18-fold upon LHX2 KD. (G) Luciferase assay for comparing the effect of the 2 alleles of rs3809579 on the activity of the regulatory element bound by LHX2 in the promoter of TRPM1 following LHX2 overexpression. The *p*-value calculated by *t* test, paired, two-tailed, (*N* = 6, the data and analyses are detailed in S10 Table). AI, allelic imbalance; AMD, age-related macular degeneration; CRE, *cis*-regulatory element; eQTL, expression-quantitative trait loci; KD, knockdown; RPE, retinal pigmented epithelium; SNP, single-nucleotide polymorphism.

variant that affects AMD risk by altering TRPM1 expression in RPE through modulation of LHX2 binding to its promoter.

## Discussion

A major challenge in human genome research is to uncover the functional significance of genetic variation within noncoding regions for predisposition to complex diseases. To approach this challenge, it is essential to identify tissue-specific transcriptional regulators, delineate their cistromes, and evaluate their impact on transcriptional regulation in cell types that are relevant to the disease's pathology. Here, we discovered a novel RPE transcriptional module consisting of LHX2 and OTX2, 2 important factors for central nervous system patterning and development. We show that in the RPE, the primary tissue affected in AMD, LHX2 is required for OTX2 expression and both co-bind multiple *cis*-regulatory regions controlling the transcriptional program that defines RPE identity, including genes involved in monogenic

and complex retinal diseases. The shared activity of LHX2 and OTX2 in transcriptional regulation of the RPE program is likely mediated by co-factors that control chromatin conformation and accessibility. Indeed, our proteomic analysis revealed a direct association of the chromatin looping factor LDB1, known co-factor of LHX2, with OTX2. Moreover, we report direct association of OTX2 with multiple subunits of the SWI/SNF chromatin-remodeling complex that function to enhance DNA accessibility. Finally, by intersecting the identified LHX2-OTX2 *cis*-regulatory map with published eQTL and ATAC-seq data from human RPE, we reveal a causal noncoding genetic variant for AMD and demonstrate that it acts by altering TRPM1 expression in the RPE through modulation of LHX2 binding to its promoter.

Trpm1 and Trpm3 are structurally similar members of the transient receptor potential melastatin (TRPM) ion channels that are highly expressed in the RPE [52]. Trpm1 mutations cause pigmentation defects as well as retinal disorders known as complete congenital stationary night blindness in mammals [53]. This phenotype is attributed to Trpm1 activity in the retinal ON-center bipolar cells, whereas the roles of Trpm1 or Trpm3 in RPE are not fully understood (reviewed in [52]). Importantly, both TRPM1/3 gene loci contain within their introns the genes encoding for miR-211 and miR-204, which are co-regulated with the host genes and have similar seed sequences inferring shared targets (reviewed in [52,54]. These miRNAs are highly enriched in the RPE, where they function through regulatory loops with multiple RPE genes including Mitf and Pax6 [55,56]. Moreover functional studies in vivo and in human fetal RPE documented their requirement for epithelial integrity and regulation of phagocytic/autophagy–lysosomal pathway genes consistent with documented RPE dysfunction in AMD [57–61]. Thus, the regulation of the Trpm1 locus, which includes miR211, by the LHX2is expected to alter expression of multiple RPE genes simultaneously, culminating in modulation of susceptibility to AMD.

ChIP-seq provides an unbiased approach to mapping, on a genomic scale, the TF-bound regions and to experimentally identify the combination of TFs and co-factors associated with cell-specific regulatory regions. This approach has been instrumental for determining cell type-specific *cis*-regulatory regions of Otx2, Crx, and Lhx2 in retinal progenitor and retinal lineages [38,62,63]. Here, we mapped the LHX2 and OTX2 bound sites in hES-RPE. Of the 191 genes down-regulated in *LHX2*-KD hES-RPE, 71 (37%) were associated with a LHX2-bound site based on proximity. Of these, 69 (97%) were associated with an OTX2 regulatory-bound region, and in 59 (86%) of these cases, LHX2 and OTX2 reside within the same DNA fragment of approximately 200 bp (Fig 3). For TRPM1, we identified a single site upstream of the TSS and genetic variation within it was associated with TRPM1 expression level in the RPE. In contrast, for many other RPE-regulated genes, we identified multiple distant enhancers. The OTX2 gene itself seems to be regulated by 5 CREs, 2 of which were bound by both OTX2 and LHX2, and the other 3 were bound only by OTX2. Likewise, for *COL8A1*, an ECM component linked to AMD and to corneal dystrophies [64,65], 4 sites were bound by both OTX2 and LHX2 and 3 only by OTX2. The TRPM3 gene is associated with 3 sites bound by both LHX2 and OTX2 and 7 additional regions were bound only by OTX2.

The identification of several CREs for many of the LHX2-OTX2 targets supports functional redundancy between the enhancers, which is in line with the concept of "shadow enhancers," whereby sets of enhancers regulate a common target in a partly or completely overlapping manner (reviewed by [66]). Multiple regulatory elements may be required to establish and maintain local transcriptional hubs containing the high TF concentrations required for gene expression [67]. Importantly, multiple regulatory elements also provide a safeguard against noncoding mutations, which could be deleterious for tissue maintenance as shown here for the risk-SNP for TRPM1 in AMD.

The prevalent co-occupancy of OTX2 and LHX2 on many genomic sites suggests that they cooperate to ensure the activation of tissue-specific target genes. LHX2 and OTX2 are widely expressed in developing neural structures, where they play stage and tissue-specific roles. In the developing retina, their expression is mutually exclusive, and they have different roles: LHX2 is important for maintaining the proliferation and multipotency of retinal progenitors, and at later stages, is required for mediating the stress response of Muller glia cells [20,22,25]. In contrast, OTX2 is detected in the precursors of photoreceptors and bipolar cells and is important for the differentiation of both of these neuronal lineages [68–71]. So far, the RPE is the only eye lineage in which these 2 factors are known to be co-expressed. Our finding that most of the LHX2 target genes in the RPE are regulated by enhancers that are co-bound by LHX2 and OTX2 suggests that their combined activity is pivotal for the RPE-specific *cis*-regulatory site choices. In support of this notion, loss of OTX proteins from the RPE results in its transdifferentiation to neural tissue with retinal features [12]. Mechanistically, it could be that it is the observed reduction in OTX activity in mouse mutants causes this transdiffrentiation, due to an alteration in LHX2 *cis*-regulatory target selectivity, shifting from RPE-associated genes to ones controlling retinal neurogenesis.

The regulatory regions bound by LHX2 and OTX2 are located far from TSS. It is, therefore, expected that these regions become associated with promoter regions through the formation of chromatin loops. LDB1, a known obligatory co-factor of LIM domain proteins, is an important factor for enhancer–promoter looping in different species and lineages [72–76]. LDB1/2 do not bind the DNA directly but rather through interaction with LIM domain proteins that either contain HDs that direct the complex to specific targets or associate with other DNA-binding TFs [77,78]. Interaction between Ldb and Otx2 was previously noted in the fly embryo, where Chip, the fly homolog of Ldb1/2, was shown to physically interact in vitro with paired-type HD proteins, including Bicoid (Bcd) and Otd, the homolog of Otx2 during early segmentation of the fly embryo [79]. The interaction between Ldb1 and Otx2 has also been documented to be required during early embryonic development for anterior structures including the head in different vertebrates, including Xenopus and mice [80–82]. Here, we detect direct binding between endogenous LDB1/2 and OTX2 in hES-RPE through quantitative co-IP-MS. This, together with the detection of multiple LHX2 and OTX2 co-bound RPE distal enhancers, suggests a possible role for LDB1 in mediating the interaction between LHX2 and OTX2, in at least some of the co-bound enhancers. It suggests a role for LHX2-LDB-OTX2 in potentiating RPE-specific gene expression, possibly through the nucleation of protein complexes, local modulation of chromatin structures, and association with RPE-tissue-specific promoters.

The regulatory regions bound by LHX2 and OTX2 likely recruit chromatin modifiers and remodelers that establish and maintain tissue-specific transcription. SWI/SNF complexes are crucial for the normal development of multiple cell types and are considered to counter the repressive activity of Polycomb complexes [83–85]. The catalytic subunit Brg1 interacts directly with tissue-specific and/or growth-factor/hormone-inducible TFs (e.g., FoxA1, MyoD, Pax6, steroid hormones, and Fos/Jun (AP-1)) and plays a central role in enhancer selection [86–89]. The physical interaction of multiple subunits of SWI/SNF complexes with OTX2 in human RPE suggest a major role for OTX2 in recruiting SWI/SNF complexes to, and/or stabilizing their interaction with tissue-specific enhancers in the RPE and possibly also in other lineages of the CNS. The interactions between OTX2, LDB1, and SWI/SNF may take place with or without LHX2. Future studies are required to further characterize the proteins' composition and the physical interactions between them (OTX2 and LHX2 or OTX2 alone, with or without LDB1 and SWI/SNF) and to further determine how this impacts the transcriptional output and its relevance to retinal disease mechanisms.

## Material and methods

### Mouse lines

The *Lhx2*$^{loxP}$ [21] and *Dct-Cre* [31] mouse lines have been described previously. The genetic background of mice used in this study was C57BL/6J. Mice were maintained according to international guidelines in compliance with the American National Research Council, NRC, and their use was approved (approval number 0121013) by the Tel Aviv University IACUC review board.

### Cell culture and differentiation of hES-RPE

The 293FT and D407 cell lines were grown in Dulbecco's Modified Eagle's Medium (DMEM) supplemented with 10% Certified Foetal Bovine Serum (FBS) for 293FT and 3% for D407 cell, penicillin (100 U/ml), and streptomycin (100 μg/ml) at 37˚C and 5% $CO_2$. hES-RPE cells were generated and maintained as described [35].

### Histology and immunofluorescence

Immunofluorescence analyses and hematoxylin and eosin staining were performed on 10-μm paraffin sections as described [90]. Briefly, sections were immersed in PBS and boiled twice in Unmasking Solution (VECTOR, H3300) for antigenic determinant retrieval. The sections were blocked for 2 h in PBSTG (0.2% Tween 20, 0.2% gelatin in PBS), incubated overnight with primary antibodies, washed with PBSTG, incubated 2 h at room temperature with the secondary antibodies, washed with PBSTG, and mounted in moviol. All of the antibodies are listed in S8 Table.

hES-RPE cells were grown on 96 wells glass plate coated with 2.5 μg/ml vitronectin (GIBCO). Cells were fixed with 4% paraformaldehyde for 10 min followed by blocking and permeabilization with 0.2% Triton-X100, 5% normal goat serum, and 0.1% BSA in PBS for 1 h at RT. The cells were then stained overnight at 4˚C using primary antibodies in blocking solution, washed 3 times in PBS, and incubated with secondary antibody for 30 min. The samples were mounted in mounting medium with DAPI (GBI Labs) and images were acquired with an Olympus BX61 fluorescence microscope. Confocal images were acquired using a Nikon C2 + laser-scanning confocal microscope.

### Lentivirus production, purification, and transduction

Down-regulation of LHX2 in hES-RPE was performed using pGIPZ vectors (Thermo Fisher Scientific) for targeting LHX2 mRNA (NM_004789). The efficiency of 4 vectors was examined in D407 cells following transduction and 48 h of incubation. We examined the KD efficiency by quantitative PCR. The 2 vectors that showed the strongest effect, of 75% (V2LHS_35479, mature antisense AATGGCAAAGTAAGACTTC; and V3LHS_319912, mature antisense AGATGCTACCGTCCTTGCT) were used for LHX2 KD in hESRPE. For the negative control, we used NS-shRNA pGIPZ vector (RHS4531).

To produce the infectious viruses, the 293FT packaging cell line was co-transfected using jetPEI (Polyplus) with lentiviral backbone plasmids shRNA-LHX2-pGIPZ or the shRNA-NS-pGIPZ, packaging plasmid pCMVΔR8.2, and envelope plasmid pVSV-G. After 72 h, the virus particles in the medium were collected and filtered through a 0.45-μm pore size filter, followed by concentration using a Vivaspin (GE Healthcare).

hES-RPE cells ($2.5 \times 10^5$) were seeded in a 24-well plate and grown for 14 days. For the infection, cells were incubated with virus-containing medium in the presence of 100 μg/ml polybrene (hexadimethrine bromide; Sigma) for 30 min at 37˚C followed by centrifugation

(30 min, 37˚C, 1,100xg). Medium was then replaced with fresh growth medium. After 24 h, 10 μg/ml puromycin (Sigma) was added for an additional 5 days followed by harvesting for RNA isolation (with RNeasy Mini Kit, QIAGEN).

## RNA-seq

For transcriptomic analysis, cDNA libraries were prepared from RNA isolated from the hES-RPE transduced with LHX2-shRNA or NS-shRNA (duplicates) using RNEASY KIT (Qiagen) and qScript cDNA Synthesis Kit (QantaBio). Samples were sequenced on Illumina Next-Seq machine, using Single-Read 60 protocol. Sequenced reads were mapped to human genome version GRCh37, using TopHat v2.0.10 [91]. Genes were identified using annotations from Ensembl release 82. Per gene reads were counted using *featureCounts*. Normalization of read counts and *p*-values for DEGs were computed using DESeq2 [92]. The *p*-values were adjusted for multiple testing using Benjamini–Hochberg false discovery rate (FDR) correction. Functional enrichment analyses were conducted through Expender using TANGO [93–95]. Gene set enrichment analysis was done using GSEA software (http://www.broad.mit.edu/gsea/) [105].

## ChIP-seq

ChIP was conducted as previously described [96] on differentiated hES-RPE cells [35]. Chromatin was prepared from 10^7 cells and 20 μg of chromatin was immunoprecipitated with 5 μg of goat anti-LHX2 antibody (Millipore, **ABE1402**) or OTX2 antibody (Abcam, ab21990). ChIP-seq libraries were prepared in duplicates and sequenced using G-INCPM in-house protocol. For ChIP-seq data analysis, sequenced reads were mapped to the human genome (v19) using bowtie2 [97] (see S7 Table for alignment statistics). LHX2 and OTX2 binding sites ("peaks") were called using MACS2 [98]. We applied a permutation-based test to examine the significance of the observed overlap between the 1,278 LHX2 and 6,461 OTX2-binding sites as follows: We generated 1,000 random sets of 1,278 genomic regions that (1) fall within open chromatin in RPE cells (as determined by ATAC-seq [51]); (2) have a matched profile of GC content; and (3) distance-to-nearest-promoter distribution as the true set of 1,278 LHX2 peaks. Then, for each random set, we examined the overlap with the set of the genuine 6,461 OTX2 peaks and used this number to generate a null distribution for the overlap between the LHX2 and OTX2 peak sets. The average overlap of these random permutations with the OTX2 set was 105 peaks, markedly lower than the 864 overlapping sites observed for the true set of LHX2 peaks (Fig 3D).

## Immunoprecipitation and mass spectrometry analysis

Nuclei of approximately $2 \times 10^7$ hES-RPE cells (d14 to d30 in culture) were isolated by incubation in cytoplasmic extract buffer (10 mM HEPES, 60 mM KCl, 1 mM EDTA, 1 mM DTT with 0.015% NP40) for 10 min, 4˚C. The nuclei were lysed with 500 μl lysis buffer (1 mM $MgCl_2$, 1% protease inhibitor, 1% NP40, 1% benzonase, 150 mM NaCl, 50 mM Tris (pH 7.5)). Preclearing by incubation with protein-A agarose beads (Millipore) was followed by centrifugation and overnight incubation at 4˚C of the supernatants with 10 μg rabbit anti-OTX2 antibody (Abcam-ab21990) or anti-Lhx2/LH2 antibody (Abcam-ab184337). As a control, the lysates were precipitated with normal rabbit control IgG (Jackson, 011-000-003). Then, beads were added for an additional 2 h. Samples were washed 4 times with basic buffer (150 mM NaCl, 50 mM Tris (pH 7.5).

For mass spectrometry, the proteins immunoprecipitated by the anti-OTX2or LHX2 antibodies, from 3 repeats, were run on 10% gel and the proteins were extracted from the gel. The

proteins in the gel were reduced with 3 mM DTT in 100 mM ammonium bicarbonate (60˚C for 30 min), modified with 10 mM iodoacetamide in 100 mM ammonium bicarbonate (in the dark, room temperature for 30 min), and digested in 10% acetonitrile and 10 mM ammonium bicarbonate with modified trypsin (Promega) at a 1:10 enzyme-to-substrate ratio, overnight at 37˚C. The resultant peptides were desalted using C18 tips (Top tip, Glygen) and were analyzed by LC-MS-MS. The peptides were resolved by reverse-phase chromatography on 0.075-mm × 180-mm fused silica capillaries (J&W) packed with Reprosil reversed phase material (Dr. Maisch GmbH, Germany). They were eluted with linear 60 min gradient of 5% to 28% 15 min gradient of 28% to 95% and 15 min at 95% acetonitrile with 0.1% formic acid in water at flow rates of 0.15 μl/min. MS was performed in a Q Exactive plus mass spectrometer (Thermo) in a positive mode using repetitively full MS scan followed by collision-induces dissociation (HCD) of the 10 most dominant ions selected from the first MS scan. The raw mass spectrometry data are available via ProteomeXchange with identifier PXD038485.

The MS data were analyzed using MaxQuant software 1.5.2.8 [99] for peak picking and identification using the Andromeda search engine, versus the human UniProt database with mass tolerance of 6 ppm. Oxidation on methionine and acetylation at the N-terminus were accepted as variable modifications and carbamidomethyl on cysteine was accepted as static modifications. Minimal peptide length was set to 6 amino acids and a maximum of 2 miscleavages was allowed. Peptide- and protein-level FDRs were filtered to 1% using the target-decoy strategy. The protein table was filtered to eliminate the identifications from the reverse database and from common contaminants. The data were quantified by label-free analysis using the same software, based on extracted ion currents (XICs) of peptides enabling quantitation from each LC/MS run for each peptide identified in any of experiments. Missing values were replaced with the minimal intensities identified in the project. The protein quantifications, classifications, and statistical analyses (two-tailed, paired, *t* test) were done using Perseus 1.6.10.43 software [100].

## Overexpression and Co-immunoprecipitation

Overexpression was achieved by pcDNA3.1 expression vectors containing human OTX2 or LHX2 [15,101] or Ldb1-FLAG (in pDEST-Flag backbone; [102]).

The expression vectors were transfected in the indicated combinations into HEK-293T cells in 10-cm dishes; transfections was of 15 μg total DNA using calcium phosphate precipitation; 72 h after transfection, cells were lysed in 1 ml lysis buffer (50 mM Tris-HCl (pH 7.9), 100 mM NaCl, 0.5 mM EDTA, 2% glycerol 99%) for 10 min, 4˚C, followed by sonication (12 cycles of 30 s on, 30 s off), incubation with 1% benzonase, 1 mM $MgCl_2$ for 2 h, and centrifugation. Extracts were precleared using protein A/G plus-agarose (Santa Cruz Biotechnology SC-2003) and immunoprecipitated with anti-Flag M2 agarose (Sigma A2220). The proteins were detected by western blotting using rabbit anti-OTX2 antibody (Abcam-ab21990). Co-IP of OTX2 with LDB1-Flag was observed in 3 independent experiments.

## Luciferase reporter assay

The D407 cells ($5 \times 10^5$) were cultured in a 24-well plate for 5 h, transfected with 400 ng of empty pGL4.23 (Promega) or CRE in pGL4.23 together with LHX2, OTX2, or both cloned in pcDNA3.1 [16,101]. The total amount of expression vectors was adjusted to 1.2 μg by adding pcDNA3.1, along with 20 ng of the pRL-TK-Renilla vector, to normalize for transfection efficiency. Each data point represents average of at least 2 technical replicates (S10 Table). The number of biological replicates is indicated in the figure legends. Transfection was performed using jetPEI (Polyplus) transfection reagent. After 48 h, enhancer activity was measured using

the Dual-Luciferase Reporter Assay (Promega) according to the manufacturer's instructions. The statistical analyses were performed using GraphPad Prism 8.0 statistical software as detailed in the figure legends.

## Analysis of GWAS SNPs in LHX2 and OTX2 bound genomic loci

We first expanded the set of 15 tag SNPs (colocalized eQTL-AMD GWAS SNPs as detected by Orozco and colleagues [48] by inclusion of their LD mates ($r^2 > 0.7$) [PLINK, v1.90], generating a union of 304 candidate causal SNPs. Then, we checked for overlaps (using BEDTools, v2.27) between these AMD SNPs and the peaks of LHX2 or OTX2 in hES-RPE, identified by our ChIP-seq analysis. Motif mapping was done using FIMO [103]. Macular RPE ATAC-seq data were aligned to the human genome (GRCh37) using Rsubread (v2.0.0). In the AI analysis, to avoid bias toward the allele of the reference genome, we masked the rs3809579 base in the FASTA file with N using BEDTools (maskFastaFromBed). We used SAMTools to filter out unmatched-pair reads as well as reads with low mapping quality (MAPQ score <10) and removed duplicate reads using Picard tools (MarkDuplicates). We then ran a binomial test for the reads that cover this SNP, considering ATAC-seq samples of the 2 heterozygous individuals. We combined the $p$-values by Fisher's method (metap R package).

## Supporting information

**S1 Fig. Neural markers are not elevated in the LHX2 conditional mutation in the mouse RPE.** In E16.5 control (A, B; A', A", and B' are the respective separate channels) and $Lhx2^{loxP/loxP};DCT\text{-}Cre$ (C, D; C', C", D' are the respective separate channels) developing RPE the expression of neuronal markers Tubb3, Vsx2 (A, C), and NF-165 (B, D) are not detected by indirect immunofluorescent analyses. DAPI was used for counterstaining of the nuclei. Scale bar is 50 μm, lower insets are 10 μm.
(TIF)

**S2 Fig. hES-RPE differentiation and phenotype of LHX2 knockdown.** Expression of tight junction protein ZO-1 (green) and the transcription regulator MITF (red) in (A) d5 and (B) d14 culture of hES-RPE. Scale bar is 10μm. (C) Gene set enrichment analysis comparing gene expression profiles in differentiated (d14) and de-differentiated (d5) hES-RPE cells shows a significant enrichment for the "RPE gene signature" among the genes up-regulated in d14. (D) The lentiviral transduction of control NS-shRNA or (E) LHX2-shRNA KD to hES-RPE (d14). GFP (green) marks the transduced cells. The indirect immunofluorescent analyses were conducted with antibodies against LHX2 (red, top row), CDH1 (blue, middle row), and OTX2 (purple, lower row). Composite shown on right; adjacent are the separate channels. DAPI (blue) labels the nuclei. Scale bar is 10 μm.
(TIF)

**S3 Fig. LHX2 and OTX2 activate together the OTX2-CRE.** Luciferase reporter activity induced by LHX2 or OTX2 or both on the PGL4.23-OTX2-CRE as compared to the TFs activity on empty PGL4.23. The $p$-values calculated by $t$ test, two-tailed, paired ($N = 7$, the data and analyses are detailed in S10 Table).
(TIF)

**S4 Fig. LHX2 transcriptional activation of rs3809579.** Luciferase reporter activity induced by LHX2 overexpression on 309 bp that contain regulatory element bound by LHX2 in rs3809579. Both CRE-TRPM1-T and CRE-TRPM1-C alleles are significantly activated by LHX2 as compared to its activation of empty PGL4.23. The $p$-values calculated by one-way ANOVA, matched samples, corrected with Dunnett's for multiple comparisons ($N = 6$). In all

6 independent experiments, transcriptional activity of the CRE-TRPM1-C is higher than CRE-TRPM1-T (overall *p*-value of this consistent trend is $0.5^6 = 0.016$; binomial test). The data underlying this figure are detailed in S10 Table.
(TIF)

**S1 Raw image. Raw image of the western blot analysis presented in Fig 5B.** The 293T cells overexpressing OTX2, LDB1, and LHX2 in the indicated combinations (Input) were subjected to immunoprecipitation with Flag antibodies followed by western blot analysis and immunolabeling with OTX2 antibody.
(PDF)

**S1 Table. Enrichment of RPE signature genes [37] in differentiated (d14) hES-RPE based on gene-set enrichment analysis comparing expression profiles between d5 and d14 hES-RPE [33].**
(XLSX)

**S2 Table. Differentially expressed genes following LHX2 KD in hES-RPE S2A - lhx2_kd_hrpe_rnaseq_cnts_qnorm:_Transcriptional profile of hES-RPE following lentiviral transduction of LHX2-shRNA or non-silencing-shRNA control conducted in 2 biological repeats and analyzed by DESeq2.** The table includes fold change, the statistical analysis, and the LHX2 ChIP-seq peaks associated with the indicated genes. S2B - OTX2_28_-target_set_Fig 3A: presents the statistical analysis and fold change following LHX2-KD of the genes shown to be down-regulated following Otx2 inactivation in the mouse RPE (based on [14]). S2C - TRPM1 exp—Fig 6F: the expression of TRPM1 in hES-RPE, based on DESeq2, following lentiviral transduction of LHX2- shRNA or non-silencing-shRNA control conducted in 2 biological repeats and normalized to 1.
(XLSX)

**S3 Table. The gene ontology enrichment list for genes differentially expressed following LHX2 –KD (TANGO).**
(XLSX)

**S4 Table. The LHX2 ChIP-seq peaks in hES-RPE (d14).** The table includes the information on chromosomal coordinates (human genome (v19), peak intensity, the nearest gene, and its distance from the peak summit.
(XLSX)

**S5 Table. The OTX2 ChIP-seq peaks in hES-RPE (d14).**
(XLSX)

**S6 Table. Overlap between LHX2 and OTX2 ChIP-seq peaks.**
(XLSX)

**S7 Table. Alignment summary statistics.**
(XLSX)

**S8 Table. Antibodies used in this study for immunostaining.**
(DOCX)

**S9 Table. Interacting proteins with LHX2, OTX2, BRG1 based on immunoprecipitation followed by mass spectrometry.** The table lists the intensities (Log2) of peptides—identified in the IP-MS interactome analysis. Each row contains the name of proteins that could be reconstructed from a set of identified peptides. The MS data analysis was done by the Max-Quant software. The protein Intensities are the Summed up extracted ion current of all

isotopic clusters associated with the identified amino acid sequence. *T* tests analyses (two-tailed, paired done using Perseus software) was done between the proteins that were pulled down using specific antibodies (LHX2, BRG1, OTX2) compared to pulled down using nonspecific IgG.
(XLSX)

**S10 Table. The luciferase reporter assays including raw data, normalizations, and statistical analyses.** The table includes 2 tabs: CRE_OTX2_Fig 4_S3: the Luciferase activity of the CRE_OTX2 following overexpression of either LHX2 or OTX2 or co-expression of both. TRPM1C-T_ Fig 6_S4: the Luciferase activity of the CRE_TRPM1-T or CRE_TRPM1-C following overexpression of LHX2.
(XLSX)

## Acknowledgments

We thank Tsadok Cohen and Heiner Westphal for the Lhx2 cKO mice, Ivana Savic-Azoulay for help with tissue culture and mouse analyses, and Noriko Esumi for OTX2 and LHX2 expression vectors. Dr. David. E. Fisher and Carmit Levy for the MITF antibody. Yoni Haitin for help with tissue culture protocols.

## Author Contributions

**Conceptualization:** Mazal Cohen-Gulkar, Ahuvit David, Naama Messika-Gold, Mai Eshel, Ran Elkon, Ruth Ashery-Padan.

**Data curation:** Mazal Cohen-Gulkar, Naama Messika-Gold, Mai Eshel, Ran Elkon.

**Formal analysis:** Mazal Cohen-Gulkar, Ahuvit David, Naama Messika-Gold, Mai Eshel, Shai Ovadia, Nitay Zuk-Bar, Tamar Ziv, Ran Elkon, Ruth Ashery-Padan.

**Funding acquisition:** Ruth Ashery-Padan.

**Investigation:** Mazal Cohen-Gulkar, Ahuvit David, Naama Messika-Gold, Mai Eshel, Shai Ovadia, Nitay Zuk-Bar, Yamit Cohen-Tayar, Ran Elkon, Ruth Ashery-Padan.

**Methodology:** Mazal Cohen-Gulkar, Naama Messika-Gold, Shai Ovadia, Nitay Zuk-Bar, Maria Idelson, Benjamin Reubinoff, Tamar Ziv, Meir Shamay, Ran Elkon, Ruth Ashery-Padan.

**Resources:** Maria Idelson, Yamit Cohen-Tayar, Benjamin Reubinoff, Meir Shamay, Ran Elkon.

**Supervision:** Ran Elkon, Ruth Ashery-Padan.

**Validation:** Mazal Cohen-Gulkar, Nitay Zuk-Bar.

**Visualization:** Mazal Cohen-Gulkar, Mai Eshel, Ruth Ashery-Padan.

**Writing – original draft:** Ran Elkon, Ruth Ashery-Padan.

**Writing – review & editing:** Mazal Cohen-Gulkar, Ahuvit David, Naama Messika-Gold, Mai Eshel, Shai Ovadia, Yamit Cohen-Tayar, Tamar Ziv, Meir Shamay.

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
