## [Editor Report · Decision Letter 0]

28 Apr 2022

Dear Dr Ashery Padan, 

Thank you for submitting your manuscript entitled "The LHX2-OTX2 transcriptional-complex controls RPE differentiation and underlies a causal regulatory risk-SNP in age-related macular degeneration" for consideration as a Research Article by PLOS Biology.

I apologize again for our delay in sending you an initial decision. I have now had a chance to discuss your study with an Academic Editor and with the PLOS Biology editorial staff and I am writing to let you know that we would like to send your submission out for external peer review.

Once your full submission is complete, your paper will undergo a series of checks in preparation for peer review. Once your manuscript has passed the checks it will be sent out for review. To provide the metadata for your submission, please Login to Editorial Manager (https://www.editorialmanager.com/pbiology) within two working days, i.e. by May 02 2022 11:59PM.

If your manuscript has been previously reviewed at another journal, PLOS Biology is willing to work with those reviews in order to avoid re-starting the process. Submission of the previous reviews is entirely optional and our ability to use them effectively will depend on the willingness of the previous journal to confirm the content of the reports and share the reviewer identities. Please note that we reserve the right to invite additional reviewers if we consider that additional/independent reviewers are needed, although we aim to avoid this as far as possible. In our experience, working with previous reviews does save time. 

If you would like to send previous reviewer reports to us, please email me at lsmith@plos.org to let me know, including the name of the previous journal and the manuscript ID the study was given, as well as attaching a point-by-point response to reviewers that details how you have or plan to address the reviewers' concerns. 

Kind regards,

Luke

Lucas Smith

Associate Editor

PLOS Biology

lsmith@plos.org

---

## [Decision Letter · Decision Letter 1]

17 Jun 2022

Dear Ruth,

Thank you again for your patience while your manuscript "The LHX2-OTX2 transcriptional-complex controls RPE differentiation and underlies a causal regulatory risk-SNP in age-related macular degeneration" was peer-reviewed at PLOS Biology. Your study has now been evaluated by the PLOS Biology editors, an Academic Editor with relevant expertise, and by several independent reviewers. 

In light of the reviews, which you will find at the end of this email, we would like to invite you to revise the work to thoroughly address the reviewers' reports. As you will see from their comments, all the reviewers felt that this was an interesting study. However, the reviewers have also raised a number of important concerns, including that some of the conclusions made are not yet adequately supported by the data. In particular, the reviewers highlight that additional data is needed to show the causal link between the single polymorphism to TRPM1 expression, and the presence of OTX2 and LHX2 within the same complex.

Given the extent of revision needed, we cannot make a decision about publication until we have seen the revised manuscript and your response to the reviewers' comments. Your revised manuscript is likely to be sent for further evaluation by all or a subset of the reviewers.

**IMPORTANT - SUBMITTING YOUR REVISION**

*Re-submission Checklist*

*Published Peer Review*

*PLOS Data Policy*

*Blot and Gel Data Policy*

Sincerely,

Lucas

Lucas Smith, Ph.D.

Associate Editor

PLOS Biology

lsmith@plos.org

REVIEWS:

Reviewer #1: This is a very interesting paper dealing with transcriptional regulatory networks important for the retinal pigment epithelium of the eye. 

The authors combined mouse in vivo data with human RPE datasets for RPE regulatory regions and defined an interplany between LHX2 and OTX2.

Matching this LHX2-OTX2 cistrome (ChIP-Seq) with ATAC-seq data for open chromatin regions and ultimately with GWAS SNP data for age-related macular degeneration, the authors make a clear point that the regulatory region of the AMD risk gene TRPM1 seems to be under risk SNP-dependent regulatory control.

The group invested a lot of work and did a great job in applying sophisticated methods important for regulatorxy network analysis.

I have only a few minor points of criticism that would balance the paper´s scope.

-The RPE is in a tight connection with the underlying retina. Both LHX2 and other factors including Crx have been well studied in transcriptomic analyses e.g. using ChIP-Seq. Relevant papers, e.g. the first ChIP-Seq study on retinal cells (Corbo et al, 2010) should be discussed.

-The in vivo mouse model for LHX2 (Figure 1) needs a better explanation why this promoter was used for cell specific KO and also a bit more background on LHX2 in general.

-I could not find that the newly generated large scale sequencing datasets were deposited in public databases.

Reviewer #2: The manuscript studied the gene regulatory networks centering the transcription factor LHX2 in retinal pigment epithelium (RPE) cells, and used the information to uncover causal non-coding regulatory risk SNPs for AMD. The authors first showed that LHX2 is required for the maintenance of RPE genes in mice RPE in vivo and in differentiated human RPE. They then found that LHX2 and OTX2 co-occupied on cis-regulatory elements and form a complex with chromatin remodeling factors to collectively regulate RPE gene expression. Lastly, they identified an AMD risk SNP that affects TRPM1 regulation by the LHX2-OTX2 complex by intersecting published ATAC-seq, GWAS and genomic datasets. 

The manuscript is of interest to readers in the field of genetics and genomics who study gene regulatory networks and their association with diseases. While the primary claims are well supported by the data, additional investigation and explanation are necessary to make the study more convincing. 

Major concerns:

1. The manuscript mostly focused on the role of Lhx2 in maintaining RPE gene expression (in differentiated human RPE). But the first part of the results focused on Lhx2's function in specifying RPE. This seems to be a little disconnected. Have the authors tried to delete Lhx2 in mature RPE in mouse in vivo and assess its function in maintaining RPE genes?

2. The authors claimed that LHX2 does not directly interact with OTX2, but forms a complex with OTX2 via LDB1. However, it is also possible that OTX2 and LHX2 form complexes with LDB1 independently. The OTX2/LDB1 and LHX2/LDB1 complexes bind to neighboring regions in the genome without real physical interactions. The authors need to provide additional information if they claim that OTX2 and LHX2 are in the same complex. 

3. Combining LHX2/OTX2 ChIP-seq data with available GWAS and ATAC-seq datasets to identify non-coding risk variants is very interesting and could have wider impact in the field. However, the author didn't verify their findings experimentally. For example, would LHX2 activate a reporter construct containing the TRPM1 variant in differentiated human RPE?

4. RNA-seq results were not verified by qPCR, RNA in situ or protein staining. 

Reviewer #3: The authors describe a set of experiments aimed at studying the role of transcription factors LHX2 and OTX2 in retinal pigment epithelium development and AMD disease development. They use transcriptome profiling in a conditional Lhx2 KO mouse model and Knockdown protocol in well-defined RPE model of differentiated human embryonic stem cells (hES-RPE). Chip-seq experiments in the hES-RPE cells are described to identify LHX2 and OTX2 genomic binding sites. From the observed regulation of OTX2 expression by LHX2 and the spatial association of the LHX2 and OTX2 putative genomic binding sites, the authors suggest that these TFs function jointly in the RPE. Since they are unable to demonstrate direct interaction between the two TFs they suggest that they both bind a third factor, LDB1. They also identify a single nucleotide polymorphic (SNP) that was described as an expression-quantitative trait locus (eQTL) that changes a putative LHX2 binding motif with a LHX2/OTX2 ChIP-seq defined binding site (peak). Based on analyses of existing RPE ATAC-seq data they conclude that the base change in the SNP change the LHX2 affinity to the locus which results in reduced expression of the target gene, TRPM1, and increased AMD risk.

This manuscript describes a significant amount of very interesting work. However, it includes several deficits that should be corrected before publication. Specifically some of the experiments are poorly described, the figure legends are poorly written and some of the conclusions are not well supported by the data provided. 

Below are some comments that may help improve the manuscript.

1. Page 13, last sentence of the first paragraph. Where is the data supporting higher affinity of LHX2 for the putative binding site with the C allele? Please describe the experiment that show that the C allele is associated with higher expression level of TRPM1 transcripts. Is this data from human RPE cells?

2. The association of the increased AMD risk eQTL tag SNP rs3809579 with the ChIP-seq and ATAC-seq signals is interesting. However, very little data is provided to support a causal roll for this variant in the regulation of TRPM1 expression. The data includes 2 ATAC-seq samples with very few reads. To substantiate this observation the authors should introduce the C allele into the hES-RPE cells model, that will allow for direct assessment of the contribution of the C allele to LHX2 binding affinity, TRPM1 expression, and chromosome accessibility.

3. The analysis of the ATAC-seq data is lacking. For example, have the authors removed duplicated fragments form the analysis?

Figure3:

1. Explain the schematic in figure 3A. What data was used to create this figure? What is the meaning of the different symbols? The description of this experiment in the result section is not clear.

2. Explain the data presented in figure 3C. This should probably only be mentioned in the results section. What were the results for the LHX2 ChIP-seq peaks?

3. Explain why the OTX2 motifs presented in figure 2J, figure 3C, and figure 3F are different. Similarly, the3 LHX2 motifs in figure 2J, figure 3F and figure 5C are different.

4. Description of panel F figure 3 is missing.

Figure 4:

1. Please define "tars" in figure 4a.

2. A basic explanation of the network examination using ingenuity pathway analysis is missing from the results section and from the figure legend.

3. The western blot in figure 3 D is of low quality. The authors should provide a better-quality blot to support direct interaction between LBD1 and OTX1. The high background in the OTX2 and LBD1 IP lanes should be addressed. What is the rational of adding LHX2 to the experiment?

Figure 5:

1. Is the data presented in panel A from this study? If not, it should be referred to in the results or discussion sections with reference, not reproduced in a major figure.

2. In B, please explain the various boxes. What exactly are ATAC-seq D1 and ATAC-seq D2?

3. The schematics in C and D seem excessive.

Reviewer #4: The manuscript of Cohen et al reports the genomic occupancy and functional role of the Lhx2 and Otx2 transcription factors in retinal pigmented epithelium. This data led to the hypothesis that a previously identified genetic variant associated with age-related macular degeneration is causal due to its decreased Lhx2 occupancy. Overall this work presents an exciting story with advances made in basic biology but additionally with a link from that data to a prevalent human disease. A strength is found in the diverse set of techniques used to generate the data to build the case for the hypothesis. However, there are further experiments that are required to fully support the conclusions that the manuscript leads to. The most important issue is to provide experimental evidence that the identified polymorphism does in fact affect Lhx2 binding and transcriptional activity.

Major

The link between the genomic occupancy of Lhx2 and the previously published eQTL data has several correlation links, but would be greatly strengthened by more empirical support to strengthen the disease-relevant conclusion. While the "C" in the first nucleotide position of the Lhx2 motif is certainly more represented in the Lhx2 consensus motif (note that the authors need to provide specifics on where this motif is from - a previous study, a particular cell type from ChIP, Selex, etc), it is not hugely overrepresented and a "T" (the alternative allele) is found at this position. The authors use the correlation between the presence of the T allele and decreased expression of TRPM1 in Fig 5A to support the causal link between this allele and TRPM1 expression. But as this particular polymorphism is closely linked between other polymorphisms in the region it is not clear if this correlation differentiates it from the other nearby polymorphisms? That is, if a similar plot as shown in Fig5A was made for all of the polymorphisms in the region, would the rs3809579 polymorphism that they focus on stand out as better correlated with transcription than those other polymorphisms? 

The authors then use ATAC-Seq data from heterozygous individuals carrying the two alleles to show that the T allele is less represented than the C allele in the sequencing. This would be expected if this allele were bound less effectively by Lhx2. But it seems possible that it could also be because there are TRPM1 enhancers with variants that are more distally located and therefore the promoter region where the rs3809579 polymorphism might reflect the general transcription differences (with lower ATAC-Seq peaks at promoter regions perhaps correlating with lower transcription). Furthermore, though the authors illustrate in Fig 5D that ChIP-Seq could be used to look at Lhx2 binding between the two variants, the authors have not done this (this should be removed from the schematic). Performing this experiment would provide more direct evidence of a causal link to TRPM1 and would strengthen the conclusion. Secondly, it would be ideal to show using reporter assays that this region with this single polymorphism change has different transcriptional activity. This would be the prediction based on the manuscript's model and reporter assays allow for the effect of the polymorphism of interest to be directly tested. The ideal cell type would be an RPE cell population.

In Fig. 4D, this experiment should have negative controls to support the specificity of the interactions. One control is included which is the IP with the Flag antibody on samples only transfected with Otx2 - however as shown in Fig 4D, the Otx2 lane has a lot of background and it is not clear what the result is for this sample. The second control would be to use a control antibody for the IP. Lastly, information about the repeatability of this result is not provided. 

Minor concerns

Fig 4 - Why is there no interaction detected between Lhx2 and Otx2 through Ldb1? This could use some discussion and also should be tested in their IP experiments (IP with Otx2 in the presence of Ldb1 and Lhx2 and probe for Lhx2 for example - and/or Lhx2 IP and Otx2 probe).

Pg3 "…as for uncovering mechanism…" needs to be rephrased

Pg 4 "Among these the Orthodenticle…" Incomplete sentence?

Pg 4 "Otx2 is also maintained in the RPE in…where it required…" needs to be rephrased

Pg 4 - "…Lim HD TF Lhx2, the orthologue…" Ortholog or paralog?

Pg 5 - a preview of the results to be shown in the paper could be useful here.

Dapi signal of E16.5 (J) - not very clear image

Pg9 "…two enhancers associated with the OTX2 gene…" Is there evidence to suggest that they are indeed enhancers? That is have they been tested for reporter activity in one of the many reports of cis-regulatory elements in the Otx2 region?

Pg 10 "… by which LHX2 activate gene expression…" needs to be rephrased

Pg 11 "… - a known co-factor of LDB1 complexes…" no reference for this statement

Pg 12 "… and that their activity, specifically that of Otx2 - …" Not sure what this phrase (and the whole sentence together) means - it needs to be clarified and referenced if there is a reference.

Pg 12 "…from numerous distal RPE enhancers that control the RPE specific transcriptional output" If this refers to the data that is only from this paper this seems to not be well supported. There is no evidence that these elements drive specific transcription or that they aren't also active in other cell types or that the BAF complex is critical for some or any activity. 

Pg 13 "This SNP is very common…" It seems most likely this refers to the T allele, but this should be stated more clearly.

Pg 15 "…that if acts by altering TRPM1…" "it" instead of "if"

Pg 15 "Thus, the regulation of the Trpm1 locos…" should be "locus"

Pg. 17 - "…reveals an important role for LDB1 in mediating the interaction between LHX2 and OTX2" This statement is not formally supported as there is no data to support that they are in fact in a complex. Could it be that they bind in a mutually exclusive manner (to the same domain of LDB1 for example)?

Pg 18 "There could be several complexes, and future studies…" At some point, in the discussion perhaps, it would be good to discuss the possibility of Crx as occupying some of those "Otx2 sites" Crx has been identified to be expressed in the RPE - this is one of the best characterizations : https://academic.oup.com/hmg/article/18/1/128/614283

Fig 5A - this data needs to be described better, even though it seems to be reproduced from a previous publication - one cannot interpret it as it is currently presented.

Pg 20 "Two vectors, which showed the strongest effect…" The metric that is being used and the assay to measure should be described.

---

## [Decision Letter · Decision Letter 2]

17 Oct 2022

Dear Dr Ashery Padan,

Thank you for your patience while we considered your revised manuscript "The LHX2-OTX2 regulatory module controls RPE differentiation and underlies a causal regulatory risk-SNP in age-related macular degeneration" for consideration as a Research Article at PLOS Biology. Your revised study has now been evaluated by the PLOS Biology editors, the Academic Editor and by Reviewer 4.

Both the Academic Editor and Reviewer 4 agree that the manuscript has been strengthened during the revision, but Reviewer 4 has raised a number of concerns with some of the new data added and the extent to which the conclusions are supported. We feel these concerns preclude publication of the manuscript in its current form, however, given our interest in the study and after discussion with the Academic Editor, we would like to give you the opportunity to address the comments from Reviewer 4, in another revised manuscript.

We expect the revision to thoroughly clarify the new experiments, as Reviewer 4 suggests, and as some additional replicates/experimental work may be needed to better support the conclusions, we have opted to provide 3 months for the revision. We may send your revised manuscript back to Reviewer 4. 

Please email us (plosbiology@plos.org) if you have any questions or concerns, or would like to request an extension.

**IMPORTANT: As you address Reviewer 4's requests, we also ask that you please address our editorial requests, which are included below my signature.** 

**SUBMITTING YOUR REVISION**

*Re-submission Checklist*

*Published Peer Review*

Sincerely,

Luke

Lucas Smith, Ph.D.

Associate Editor

PLOS Biology

lsmith@plos.org

EDITORIAL REQUESTS: 

1) ETHICS STATEMENT: Please update your ethics statement to include the full name of the IACUC/ethics committee that reviewed and approved the animal care and use protocol/permit/project license. Please also include an approval number. Please include the specific national or international regulations/guidelines to which your animal care and use protocol adhered. Please note that institutional or accreditation organization guidelines (such as AAALAC) do not meet this requirement.

2) DATA AVAILABILITY: thank you for depositing your sequencing data on GEO. In addition to this, we ask that you provide all individual quantitative observations that underlie the data summarized in the figures and results of your paper be made available in one of the following forms:

a) Supplementary files (e.g., excel). Please ensure that all data files are uploaded as 'Supporting Information' and are invariably referred to (in the manuscript, figure legends, and the Description field when uploading your files) using the following format verbatim: S1 Data, S2 Data, etc. Multiple panels of a single or even several figures can be included as multiple sheets in one excel file that is saved using exactly the following convention: S1_Data.xlsx (using an underscore).

b) Deposition in a publicly available repository. Please also provide the accession code or a reviewer link so that we may view your data before publication. 

Fig 3A,D; Fig 4D; Fig 6A,F-G; Fig S3; Fig S4;

>>Please also ensure that figure legends in your manuscript include information on where the underlying data can be found, and ensure your supplemental data file/s has a legend.

>>Please ensure that your Data Statement in the submission system accurately describes where your data can be found.

For more information on the PLOS Data Policy, see http://journals.plos.org/plosbiology/s/data-availability or this editorial: http://dx.doi.org/10.1371/journal.pbio.1001797

REVIEWS:

Reviewer #4: The revised manuscript of Cohen-Gulkar et al. has addressed many of the concerns addressed by this reviewer. One of the key experiments proposed by multiple reviewers was to test the transcriptional activity of the two variants in RPE cells to determine if the SNP was causal in creating a transcriptional difference. The authors conducted a luciferase reporter experiment in a RPE cell line and show new results in Figures S4 and 6. The presented difference for this experiment is quite modest and some of the details are unclear and need clarification (see below). Given that this is a pivotal experiment for the model and the differences in the two alleles are not large, it is reasonable to ask if the evidence is strong enough to support the conclusion that there is a difference in SNP activity.

1) The main text suggests that the results shown in Figure 6G are under conditions where Lhx2 is overexpressed. But the figure legend or the figure mark-up makes no mention of this. This should be clarified.

2) Assuming that Figure 6G and SuppF4 both use Lhx2 misexpression, why are these data plotted differently - Supp4F shows a bar graph with the (empty) basal vector data in its own column. But Figure 6G does not have this data shown and instead has the empty vector factored into the data, as suggested by the y-axis notation. How was this done? Were samples randomly paired together? Was a separate set of controls done for each? Are the datapoints shown in Figure 6G the same as those shown in SuppF4? If a statistical test was done to directly compare the T and C variant activity shown in SuppF4, would these be statistically different? If not, does this suggest that the evidence for a functional difference in Fig 6G is not solid?

3) The variability of the datapoints in SuppF4 suggests that reproducibility is an important issue. How many times has this experiment been repeated? This should be described. Can the authors provide a rationale for why they need to add Lhx2 to the mix to get this result - is there Lhx2 in these cells already, and if so, why isn't that sufficient to activate the reporter? Can reporters for the C and T alleles be introduced into the same cells so that sample variability is decreased and is instead controlled internally? Does the luciferase system allow one to use renilla to be driven by one of the variants and luciferase by the other (instead of renilla being driven by an unrelated element).

---

## [Editor Report · Decision Letter 3]

16 Nov 2022

Dear Dr Ashery Padan,

Thank you for the submission of your revised Research Article "The LHX2-OTX2 regulatory module controls RPE differentiation and underlies a causal regulatory risk-SNP in age-related macular degeneration" for publication in PLOS Biology. Your manuscript has now been evaluated by the Academic Editor, Sui Wang, and the PLOS Biology editorial team, and I am pleased to say that we are satisfied by the revision and that we can, in principle, accept your manuscript for publication, provided you address any remaining formatting and reporting issues. These will be detailed in an email you should receive within 2-3 business days from our colleagues in the journal operations team; no action is required from you until then. Please note that we will not be able to formally accept your manuscript and schedule it for publication until you have completed any requested changes.

**IMPORTANT: As you address these last formatting and reporting requests, to come, we also ask that you attend to the following editorial requests:

1) Title: After some discussion within the team, we think that the title of your piece should be edited a bit, for clarity (for example, to spell out what RPEs are). We would suggest that you change it to something like: 

"The transcription factors LHX2 and OTX2 form a regulatory module that controls retinal pigmented epithelium differentiation and underlies a risk SNP in age-related macular degeneration"

or 

"The LHX2-OTX2 transcriptional regulatory module controls retinal pigmented epithelium differentiation and underlies genetic risk for age-related macular degeneration"

2) Data: Thank you for providing the data underlying your figures as a supplementary excel file. Can you please add a sentence to each relevant figure legend referencing this table? For example, to each figure legend you could add the sentence "the data underlying this figure can be found in Table ___.

3) Ethics Statement: Please update your ethics statement to include the approval number for your animal care and use protocol, approved by the Tel Aviv University review board. Please also include the name of the specific national or international regulations/guidelines to which your animal care and use protocol adhered. Please note that institutional or accreditation organization guidelines (such as AAALAC) do not meet this requirement.

PRESS

Sincerely, 

Lucas Smith, Ph.D., Ph.D.

Associate Editor

PLOS Biology

lsmith@plos.org